



# Validation of the Aeolus L2B wind product with airborne wind lidar measurements in the polar North Atlantic region and in the tropics

Benjamin Witschas[1], Christian Lemmerz[1], Alexander Geiß[2], Oliver Lux[1], Uwe Marksteiner[1], Stephan Rahm[1], Oliver Reitebuch[1], Andreas Schäfler[1], and Fabian Weiler[1]

[1]Deutsches Zentrum für Luft- und Raumfahrt e.V. (DLR), Institut für Physik der Atmosphäre, 82234 Oberpfaffenhofen, Germany
[2]Ludwig-Maximilians-Universität, Meteorologisches Institut, 80333 München, Germany

**Correspondence:** Benjamin Witschas, (Benjamin.Witschas@dlr.de)

**Abstract.** During the first three years of European Space Agency's Aeolus mission, the German Aerospace Center (Deutsches Zentrum für Luft- und Raumfahrt, DLR) performed four airborne campaigns deploying two different Doppler wind lidars (DWL) on-board the DLR Falcon aircraft, aiming to validate the quality of the recent Aeolus Level 2B (L2B) wind data product (processor baseline 11 and 12). The first two campaigns, WindVal III (Nov/Dec 2018) and AVATAR-E (Aeolus Validation Through Airborne Lidars in Europe, May/Jun 2019) were conducted in Europe and provided first insights in the data quality at the beginning of the mission phase. The two later campaigns, AVATAR-I (Aeolus Validation Through Airborne Lidars in Iceland) and AVATAR-T (Aeolus Validation Through Airborne Lidars in the Tropics), were performed in regions of particular interest for the Aeolus validation: AVATAR-I was conducted from Keflavik, Iceland between 9 September and 1 October 2019 to sample the high wind speeds in the vicinity of the polar jet stream. AVATAR-T was carried out from Sal, Cape Verde between 6 September and 28 September 2021 to measure winds in the Saharan dust-laden African easterly jet. Altogether, 10 Aeolus underflights were performed during AVATAR-I and 11 underflights during AVATAR-T, covering about 8000 km and 11000 km along the Aeolus measurement track, respectively. Based on these collocated measurements, statistical comparisons of Aeolus data with the reference lidar (2-$\mu$m DWL) as well as with in-situ measurements by the Falcon were performed to determine the systematic and random error of Rayleigh-clear and Mie-cloudy winds that are contained in the Aeolus L2B product. It is demonstrated that the systematic error almost fulfills the mission requirement of being below $0.7\,\mathrm{m\,s^{-1}}$ for both Rayleigh-clear and Mie-cloudy winds. The random error is shown to vary between $5.5\,\mathrm{m\,s^{-1}}$ and $7.1\,\mathrm{m\,s^{-1}}$ for Rayleigh-clear winds and is thus larger than specified ($2.5\,\mathrm{m\,s^{-1}}$), whereas it is close to the specifications for Mie-cloudy winds ($2.7$ to $2.9\,\mathrm{m\,s^{-1}}$). In addition, the dependency of the systematic and random errors on the actual wind speed, the geolocation, the scattering ratio and the time difference between 2-$\mu$m DWL observation and satellite overflight is investigated and discussed. Thus, this work contributes to the characterization of the Aeolus data quality in different meteorological situations and allows to investigate wind retrieval algorithm improvements for reprocessed Aeolus data sets.





# 1   Introduction

On 22 August 2018, the first ever space-borne Doppler wind lidar Aeolus, developed by the European Space Agency (ESA), was launched into space to circle the Earth on a sun-synchronous orbit at about 320 km altitude with a repeat cycle of seven

days (e.g., ESA, 1999; Stoffelen et al., 2005; ESA, 2008; Reitebuch, 2012; Horányi et al., 2015). Since then, Aeolus has been providing profiles of the wind vector component along the instrument's line-of-sight (LOS) direction from ground up to about 30 km in the stratosphere (e.g., Kanitz et al., 2019; Reitebuch et al., 2020; Straume et al., 2020), primarily aiming to improve numerical weather prediction (NWP) (e.g., Weissmann and Cardinali, 2007; Tan et al., 2007; Marseille et al., 2008; Horányi et al., 2015; Rennie et al., 2021). Especially wind profiles acquired over the Southern Hemisphere, the tropics and the oceans

contribute to closing large gaps in the availability of wind data in the global observing system (Baker et al., 2014). For the use of Aeolus observations in NWP models, a detailed characterization of the data quality as well as the minimization of systematic errors is crucial. Thus, several scientific and technical studies have been performed and published in the meanwhile, addressing the performance of ALADIN (Atmospheric LAser Doppler INstrument) on-board Aeolus and the quality of the wind data products (e.g., Bedka et al., 2021; Martin et al., 2021; Baars et al., 2020; Guo et al., 2021; Zuo et al., 2022; Wu et al., 2022;

Chou et al., 2021; Belova et al., 2021).

The German Aerospace Center (Deutsches Zentrum für Luft- und Raumfahrt, DLR) performed four airborne calibration and validation (CalVal) campaigns since the launch of Aeolus, deploying two different Doppler wind lidars (DWL) on-board the DLR Falcon aircraft, aiming to validate the quality of the Level 2B (L2B) wind data product. During the first two campaigns, WindVal III (5 November 2018 until 5 December 2018) and AVATAR-E (6 May 2019 until 6 June 2019), that were conducted

from the DLR site in Oberpfaffenhofen, Germany, 10 satellite underflights covered more than 7500 km along the Aeolus track. The performed measurements gave first insights in the quality of the early-phase Aeolus wind product. Based on collocated measurements with the ALADIN airborne demonstrator (A2D) (Reitebuch et al., 2009) during the WindVal III campaign, a statistical comparison against Aeolus data revealed a positive systematic error of $2.6 \mathrm{~m \, s^{-1}}$ for the Aeolus Rayleigh-clear winds and a corresponding random error of $3.6 \mathrm{~m \, s^{-1}}$ (Lux et al., 2020). These results were confirmed (Witschas et al., 2020)

by a comparison with precise wind speed and wind direction measurements performed with DLR's 2-$\mu$m DWL (Witschas et al., 2017). The mean systematic errors of the Aeolus winds with respect to 2-$\mu$m DWL observations were determined to be $2.1 \mathrm{~m \, s^{-1}}$ (Rayleigh-clear) and $2.3 \mathrm{~m \, s^{-1}}$ (Mie-cloudy). The corresponding random errors were found to be $3.9 \mathrm{~m \, s^{-1}}$ and $2.0 \mathrm{~m \, s^{-1}}$. Additionally, Witschas et al. (2020) discussed results from the subsequent AVATAR-E campaign, yielding systematic errors of $-4.6 \mathrm{~m \, s^{-1}}$ (Rayleigh-clear) and $-0.2 \mathrm{~m \, s^{-1}}$ (Mie-cloudy). The corresponding random errors were $4.3 \mathrm{~m \, s^{-1}}$ and

$2.0 \mathrm{~m \, s^{-1}}$. The larger systematic errors during AVATAR-E were related to small temperature fluctuations across the 1.5 m diameter primary mirror of the Aeolus telescope which caused varying wind biases along the orbit of up to $8 \mathrm{~m \, s^{-1}}$ (Rennie and Isaksen, 2020; Rennie et al., 2021; Weiler et al., 2021b). In the meantime (since 20 April 2020), the impact of these thermal fluctuations is successfully corrected in the Aeolus processor by means of ECMWF (European Centre for Medium-Range Weather Forecasts) model-equivalent winds. Furthermore, the Aeolus detector showed anomalies in the dark current on single

pixels, which led to wind speed errors of up to $30 \mathrm{~m \, s^{-1}}$, depending on the strength of the atmospheric signal (Weiler et al.,



2021a). A corresponding correction scheme that was implemented in the Aeolus data processor on 14 June 2019 significantly reduced the impact of these hot pixels on the wind data quality.

The enhanced systematic error of Aeolus wind data in the early mission phase was further demonstrated by other data sets. For instance, NASA conducted five research flights over the Eastern Pacific Ocean with their DC-8 aircraft equipped
with a heterodyne detection Doppler wind lidar and a water vapor lidar in April 2019 (Bedka et al., 2021) which revealed a systematic error of $1.2\,\mathrm{m\,s}^{-1}$ (Rayleigh-clear) and $2.0\,\mathrm{m\,s}^{-1}$ (Mie-cloudy). The corresponding random errors were determined to be $5.1\,\mathrm{m\,s}^{-1}$ and $4.7\,\mathrm{m\,s}^{-1}$. Baars et al. (2020) used radiosonde data launched during the *Polarstern* research vessel cruise from Bremerhaven to Cape Town in Nov/Dec 2018 and determined the systematic and random error of the Rayleigh-clear winds to be $1.5\,\mathrm{m\,s}^{-1}$ and $3.3\,\mathrm{m\,s}^{-1}$, respectively. Martin et al. (2021) performed a statistical validation of Aeolus observations
using collocated radiosonde measurements and NWP forecast equivalents from two different global models, the ICOsahedral Nonhydrostatic model (ICON) of Deutscher Wetterdienst (DWD) and the ECMWF Integrated Forecast System (IFS) model, as reference data. The analysis, that covered the northern hemisphere in the time period August 2018 to December 2019, showed strong spatial variations of the Aeolus wind bias and differences between ascending and descending orbits in agreement with the aforementioned thermal fluctuations on the Aeolus telescope mirror which are different for the respective orbit direction
and were not corrected in that time frame. The mean absolute bias for the selected validation area is found to be in the range of $1.8 - 2.3\,\mathrm{m\,s}^{-1}$ (Rayleigh-clear) and $1.3 - 1.9\,\mathrm{m\,s}^{-1}$ (Mie-cloudy). In addition, radar wind profiler (RWP) measurements over China, Australia and Japan were used for Aeolus validation. In combination with ground-based lidar and radiosonde measurements a comparison of RWP measurements over Japan against Aeolus data demonstrate that the systematic error reduces significantly with improved algorithms in the Aeolus L2B data processor (from version L2B02 to L2B10) , which is
due to the implemented telescope mirror temperature correction and other improvements as for instance the correction of hot pixels on the detector (Iwai et al., 2021). In particular, for the L2B02/L2B10 period, the systematic errors were determined to be $0.5$ to $1.7\,\mathrm{m\,s}^{-1}/-0.8$ to $0.5\,\mathrm{m\,s}^{-1}$ (Rayleigh-clear) and $1.6$ to $2.4\,\mathrm{m\,s}^{-1}/-0.7$ to $0.2\,\mathrm{m\,s}^{-1}$ (Mie-cloudy), respectively. The corresponding random errors were $6.7\,\mathrm{m\,s}^{-1}/6.4\,\mathrm{m\,s}^{-1}$ (Rayleigh-clear) and $5.1\,\mathrm{m\,s}^{-1}/4.8\,\mathrm{m\,s}^{-1}$ (Mie-cloudy). The successful implementation of error corrections in the Aeolus L2B processor was also demonstrated by Guo et al. (2021) and Zuo et al.
(2022) who used RWP measurements over China from April to July 2020 and over Australia from October 2020 until March 2021, respectively, to reveal a smaller mean systematic error of $-0.6\,\mathrm{m\,s}^{-1}$ (Rayleigh-clear) and $-0.3\,\mathrm{m\,s}^{-1}$ (Mie-cloudy), or $0.7\,\mathrm{m\,s}^{-1}$ for both Rayleigh-clear and Mie-cloudy winds. Besides that, Wu et al. (2022) used ground-based heterodyne detection Doppler wind lidar measurements in the timeframe from January to December 2020 and determined systematic errors of $-1.2\,\mathrm{m\,s}^{-1}$ (Rayleigh-clear) and $-0.3\,\mathrm{m\,s}^{-1}$ (Mie-cloudy) and random errors of $5.8\,\mathrm{m\,s}^{-1}$ (Rayleigh-clear) and
$2.6\,\mathrm{m\,s}^{-1}$ (Mie-cloudy), respectively. A summary of the validation results from different CalVal campaigns is given in Table 1, containing the time-period of the respective campaigns, the L2B processor version that was operational within this time period, the systematic error $\mu$ and random error $\sigma$ of Rayleigh-clear and Mie-cloudy winds, as well as the reference instrument that was used.

It can be seen that most of the CalVal activities using observations as a reference to determine the systematic and random
errors in specific geographical regions for random wind situations above the measurement sites. In this paper, this work is



**Table 1.** Overview of Aeolus L2B validation results from different campaign data sets

| Period | Processor L2B | Rayleigh-clear | | Mie-cloudy | | Ref. instrument | Reference |
|---|---|---|---|---|---|---|---|
| | | $\mu/(\mathrm{m\,s^{-1}})$ | $\sigma/(\mathrm{m\,s^{-1}})$ | $\mu/(\mathrm{m\,s^{-1}})$ | $\sigma/(\mathrm{m\,s^{-1}})$ | | |
| Nov/Dec 18 | 02 | 2.1 | 3.9 | 2.3 | 2.0 | Airborne wind lidar | Witschas et al. (2020) |
| Nov/Dec 18 | 02 | 2.6 | 3.6 | - | - | Airborne wind lidar | Lux et al. (2020) |
| Nov/Dec 18 | 02 | 1.5 | 3.3 | - | - | Radiosondes | Baars et al. (2020) |
| Oct 18/Dec 18 | 02 | 0.5 to 1.7 | 6.7 | 1.6 to 2.4 | 5.1 | Radio wind profiler | Iwai et al. (2021) |
| Apr 19 | 02 | 1.2 | 5.1 | 2.0 | 4.7 | Airborne wind lidar | Bedka et al. (2021) |
| May/Jun 19 | 03 | 4.6 | 4.3 | -0.2 | 2.0 | Airborne wind lidar | Witschas et al. (2020) |
| Aug 18/Dec 19 | 02 to 07 | 1.8 to 2.3 | - | 1.3 to 1.9 | - | Radiosondes/models | Martin et al. (2021) |
| Jan 20/Dec 20 | 07 to 11 | 1.2 | 5.8 | -0.3 | 2.6 | Ground-based wind lidar | Wu et al. (2022) |
| Apr 20/Oct 20 | 10 | -0.8 to 0.5 | 6.4 | -0.7 to 0.2 | 4.8 | Radio wind profiler | Iwai et al. (2021) |
| Apr 20/Jul 20 | 08 to 09 | -0.6 | - | 0.3 | - | Radio wind profiler | Guo et al. (2021) |
| Oct 20/Mar 21 | 10 to 11 | 0.7 | - | 0.7 | - | Radio wind profiler | Zuo et al. (2022) |

$\mu$ denotes the systematic error and $\sigma$ the random error.

extended by the analysis of the L2B wind quality in two dedicated regions over the North Atlantic: in the extratropical polar jet stream and the tropics, whereas both regions are of particular importance to NWP (Tan and Andersson, 2005; Schäfler et al., 2020). In particular, two airborne campaigns with the Falcon aircraft being equipped with the A2D and the 2-$\mu$m DWLwere conducted, namely the AVATAR-I campaign (Keflavik, Iceland, 9 September until 1 October 2019) and the AVATAR-T campaign (Sal, Cape Verde, 6 September until 28 September 2021). The 2-$\mu$m DWL data set acquired during these two campaigns is used to derive the systematic and random error of Rayleigh-clear and Mie-cloudy winds and to investigate their dependency on different quantities as the actual wind speed, the geolocation, the scattering ratio and the time difference between 2-$\mu$m DWL observation and satellite overflight for a more detailed error characterization. In addition to the 2-$\mu$m DWL measurements, in-situ observations with the Falcon nose-boom are used for comparison. A dedicted study of the Aeolus measurement principle, its calibration procedures and retrieval algorithms is performed based on A2D observations as discussed in Lux et al. (2020) and Lux et al. (2022a).

The paper is structured as follows. In Sect. 2, an overview is given about the AVATAR-I and AVATAR-T campaign, followed by an introduction of the instruments used in this study (Sect. 3), namely ALADIN on-board Aeolus (Sect. 3.1), DLR's 2-$\mu$m DWL (Sect. 3.2) and the flow angle sensor in Falcon's nose-boom (Sect. 3.3). Afterwards, the data processing steps are discussed in Sect. 4, containing the explanation of the averaging procedures (Sect. 4.1), the introduction of the quantities used for the statistical comparison (Sect. 4.2) as well as an explanation of the quality control that is applied to the Aeolus data (Sect. 4.3). In Sect. 5, the results of the statistical comparison are discussed for the systematic errors (Sect. 5.1) as well as for the random errors (Sect. 5.2). The results retrieved from the comparison against Falcon in-situ measurements are separately treated in Sect. 5.3. In Sect. 6, the Aeolus error dependency on various quantities is revealed for Rayleigh-clear (Sect. 6.1) and Mie-cloudy winds (Sect. 6.2), followed by a summary of the results of this study given in Sect. 7.



## 2   Validation campaigns overview

In this study, results from the AVATAR-I (Aeolus Validation Through Airborne Lidars in Iceland) campaign, conducted from
9 Sept 2019 to 1 Oct 2019 from Keflavik in Iceland, and by the AVATAR-T (Aeolus Validation Through Airborne Lidars in
the Tropics) campaign, performed from 6 Sept to 28 Sept 2021 in Sal, Cape Verde, are presented. AVATAR-T was DLR's
contribution to the international Joint Aeolus Tropical Atlantic campaign (JATAC) initiated by ESA, which combined several
airborne participants as the French SAFIRE Falcon 20 and the NASA DC-8 (based on the US Virgin Islands) with a number
of ground based measurements and the deployment of a light aircraft with aerosol in-situ equipment from the University
of Nova Goriza (Slovenia), both performed from Mindelo on the island of São Vincente, Cape Verde (Fehr et al., 2021).
During both campaigns, the DLR Falcon was equipped with two well-established wind lidar systems that have already been
deployed in several Aeolus pre-launch campaigns (Marksteiner et al., 2018; Schäfler et al., 2018; Lux et al., 2018). In particular,
the Falcon hosted the A2D, which is a prototype of the ALADIN instrument with representative design and measurement
principle (Reitebuch et al., 2009). Hence, the A2D is the optimal instrument to validate the Aeolus measurement principle,
calibration procedures and retrieval algorithms. In addition to the A2D, a heterodyne detection wind lidar (2-$\mu$m DWL) with a
high sensitivity to particulate returns was flown and acted as a reference system (Witschas et al., 2017) to validate the quality
of the Aeolus wind product.

   During AVATAR-I, a total of 10 Aeolus underflights were performed, including four flights along descending orbits which
were not possible during previous campaigns (see also Fig. 1, left and Table 2). The first underflight along an ascending orbit
could already be performed on the transfer from Oberpfaffenhofen to Keflavik after a refueling stop-over in Prestwick, UK.
During the 10 underpasses, about $8000\,\mathrm{km}$ of the Aeolus measurement track were sampled by the two lidars. In contrast to the
previous CalVal campaigns, Aeolus operated with a dedicated range bin setting (RBS) that was exclusively applied in the area
around Iceland within the AVATAR-I time frame as it is shown in Fig.2, left. The setting was optimized to a higher resolution of
$500\,\mathrm{m}$ throughout the troposphere for both the Rayleigh and the Mie channel and thus provided a better overlap with more data
points between the observations of Aeolus and the airborne wind lidars at the expense of an increased noise level. Furthermore,
the high wind speeds in the vicinity of the jet stream were better resolved. Above $10\,\mathrm{km}$, the three Rayleigh range bins were
set to $1\,\mathrm{km}$ size to extend the maximum sampled altitude to about $13\,\mathrm{km}$ and with that, to assure that also winds in the lower
stratosphere were measured. During AVATAR-T, 11 Aeolus underflights could be performed, covering about $11000\,\mathrm{km}$ of
the Aeolus measurement track (see also Fig. 1, right and Table 2). Six of these flights were performed along ascending orbits
and five along descending orbits in the morning hours. During AVATAR-T, situations with Saharan dust transport together
with moderate wind cases could be targeted, making the AVATAR-T data set valuable for investigating the impact of aerosols,
especially on the Rayleigh-clear wind product. Due to the decreasing performance of Aeolus and the resulting lower signal
levels, it was not possible to apply a high-resolution RBS, as this would have led to too large noise levels. Hence, the range bin
size was kept at $500\,\mathrm{m}$ in the lower boundary layer and was increased to $750\,\mathrm{m}$ in the lower troposphere, being a compromise
between signal level and resolution of the usually aerosol-loaded SAL prominent in this altitudes. In the upper troposphere
and lower stratosphere, the range gates were set to $1\,\mathrm{km}$ (see also Fig.2, right). An overview of tracks of all flights performed





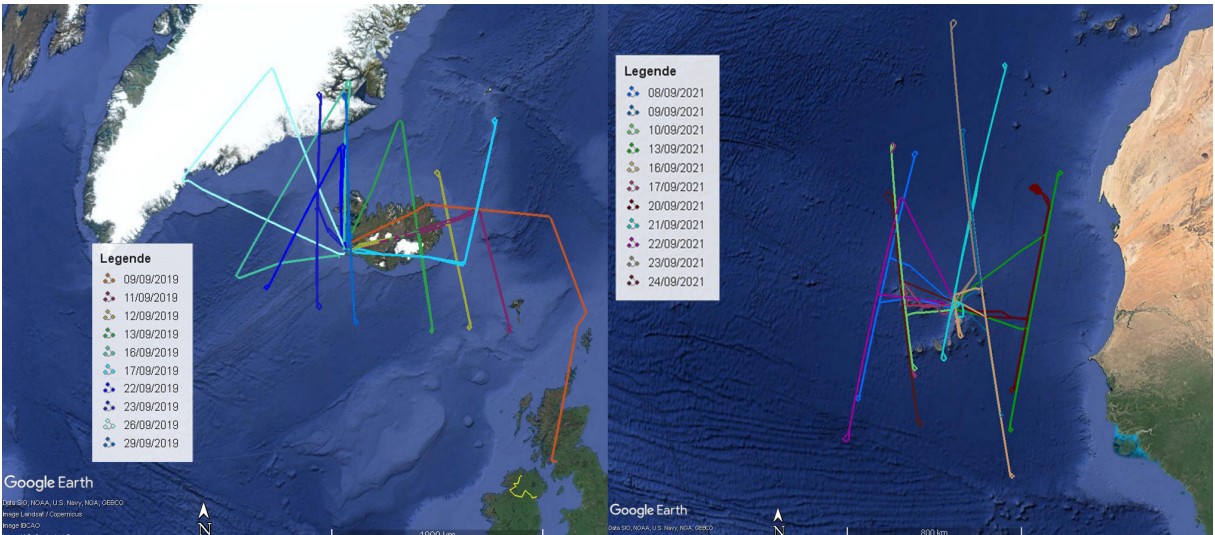

**Figure 1.** Flight tracks of the Falcon aircraft during the AVATAR-I campaign (left) and the AVATAR-T campaign (right). Each color represents a single flight.

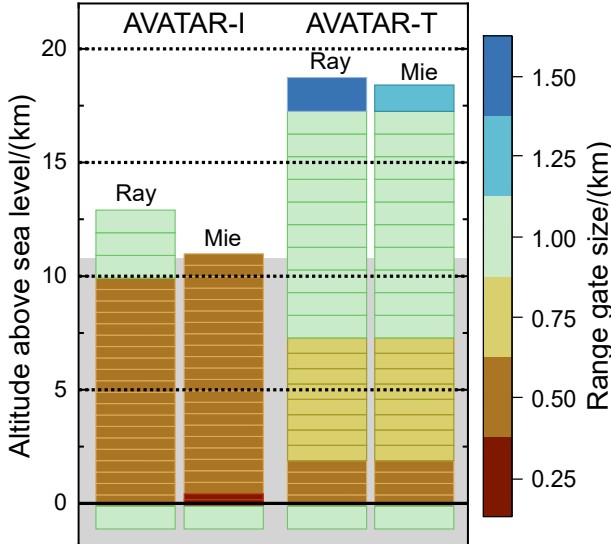

**Figure 2.** Aeolus range bin settings (RBS) for the Rayleigh and Mie channel applied during the AVATAR-I and AVATAR-T campaigns. The actual range bin size is color-coded and the gray area indicates the altitudes that are usually sampled by the airborne lidars on-board the Falcon aircraft.

during AVATAR-I and AVATAR-T is given in Fig. 1 and Table 2. The latter one also provides information about the overall duration of the research flights as well as on the start/stop times and geolocations of the Aeolus underflight, which allows for an easier access to the relevant satellite wind data for comparison. Additionally, the number of Aeolus observations that could be validated by the 2-$\mu$m DWL is given for both Rayleigh-clear and Mie cloudy winds.





**Table 2.** Overview of Aeolus underflights performed during the AVATAR-I and the AVATAR-T campaign

| | | Falcon flight | | Aeolus underflight | | # for CalVal | |
|---|---|---|---|---|---|---|---|
| | Date | Time/(UTC) | Route | Start/stop time/(UTC) | Geolocation | Ray. | Mie |
| AVATAR-I | 9/9/19 | 16:12 to 19:22 | PIK/KEF | 17:31:24 to 17:32:26 | 61.5°N/1.0°W to 65.6°N/3.1°W | 28 | 67 |
| | 11/9/19 | 16:04 to 19:28 | KEF/KEF | 17:57:19 to 17:58:36 | 61.1°N/7.4°W to 66.0°N/10.0°W | 104 | 51 |
| | 12/9/19 | 16:25 to 19:52 | KEF/KEF | 18:10:25 to 18:12:01 | 61.5°N/10.9°W to 67.6°N/14.2°W | 130 | 89 |
| | 13/9/19 | 16:52 to 20:19 | KEF/KEF | 18:23:23 to 18:25:32 | 61.2°N/14.1°W to 69.5°N/18.8°W | 154 | 79 |
| | 16/9/19 | 06:45 to 10:09 | KEF/KEF | 08:38:33 to 08:40:58 | 71.0°N/26.2°W to 61.7°N/32.0°W | 148 | 60 |
| | 17/9/19 | 05:07 to 08:43 | KEF/KEF | 07:21:01 to 07:22:32 | 69.7°N/7.8°W to 63.9°N/11.5°W | 112 | 56 |
| | 22/9/19 | 06:58 to 10:30 | KEF/KEF | 08:26:10 to 08:27:50 | 68.2°N/25.2°W to 61.7°N/28.9°W | 114 | 93 |
| | 23/9/19 | 17:38 to 21:18 | KEF/KEF | 19:02:11 to 19:04:23 | 61.5°N/24.1°W to 70.0°N/29.2°W | 108 | 177 |
| | 26/9/19 | 07:36 to 11:03 | KEF/KEF | 09:17:21 to 09:18:52 | 70.5°N/36.5°W to 64.7°N/40.5°W | 103 | 0 |
| | 29/9/19 | 17:28 to 20:47 | KEF/KEF | 18:48:59 to 18:49:57 | 61.2°N/20.7°W to 64.9°N/22.5°W | 172 | 29 |
| AVATAR-T | 8/9/21 | 05:44 to 09:28 | SID/SID | 07:39:49 to 07:42:13 | 22.5°N/25.1°W to 13.0°/26.8°W | 85 | 70 |
| | 9/9/21 | 17:25 to 21:23 | SID/SID | 19:22:20 to 19:25:08 | 12.6°N/21.0°W to 23.5°N/23.0°W | 140 | 7 |
| | 10/9/21 | 18:20 to 22:05 | SID/SID | 19:36:01 to 19:38:13 | 22.5°N/25.1°W to 13.0°N/26.8°W | 102 | 30 |
| | 13/9/21 | 05:35 to 08:18 | SID/SID | 07:14:25 to 07:16:55 | 22.0°N/18.6°W to 11.9°N/20.6°W | 94 | 13 |
| | 16/9/21 | 17:09 to 21:04 | SID/SID | 19:21:42 to 19:24:15 | 10.1°N/20.5°W to 20.3°N/22.4°W | 24 | 18 |
| | 17/9/21 | 18:06 to 21:58 | SID/SID | 19:35:33 to 19:38:13 | 13.9°N/24.6°W to 23.0°N/26.2°W | 19 | 10 |
| | 20/9/21 | 06:58 to 10:30 | SID/SID | 07:14:42 to 07:16:32 | 20.6°N/19.2°W to 13.5°N/20.5°W | 0 | 0 |
| | 21/9/21 | 05:09 to 09:12 | SID/SID | 07:26:08 to 07:29:03 | 26.4°N/21.3°W to 14.7°N/23.4°W | 10 | 3 |
| | 22/9/21 | 06:11 to 09:55 | SID/SID | 07:40:20 to 07:42:35 | 20.6°N/25.6°W to 11.7°N/27.7°W | 3 | 1 |
| | 23/9/21 | 18:05 to 21:39 | SID/SID | 19:23:42 to 19:26:10 | 18.0°N/22.2°W to 28.3°N/24.1°W | 0 | 0 |
| | 24/9/21 | 17:36 to 21:18 | SID/SID | 19:35:29 to 19:37:42 | 12.0°N/24.3°W to 21.0°N/25.9°W | 1 | 0 |

The time gives the duration between takeoff and landing. The flight route is indicated by the IATA (International Air Transport Association) airport code. PIK: Prestwick airport; KEF: Keflavik airport; SID: Amilcar Cabral airport.

## 3 Instrument overview

### 3.1 Aeolus and ALADIN

The Aeolus satellite has a weight of 1360 kg, a launch configuration dimension of 4.6 m x 1.9 m x 2.0 m and a deployable solar array that provides a power of 2.4 kW. Aeolus carries a single payload, ALADIN, which is a direct detection wind lidar operating at an ultraviolet wavelength of 354.8 nm. ALADIN emits short laser pulses ($\approx$ 40 mJ to 70 mJ, 50.5 Hz) down to the atmosphere, where a few of the photons are backscattered on air molecules, aerosols and hydrometeors. The backscattered light is collected with a 1.5 m diameter Cassegrain telescope and directed to the optical receiver that is used to detect the Doppler frequency shift of the backscattered light from which the wind velocity can be calculated in LOS direction at different altitudes. To do so, ALADIN is equipped with two different frequency discriminators, namely a Fizeau interferometer that analyzes the frequency shift of the narrow-band particulate backscatter signal by means of the so-called fringe imaging technique (McKay, 2002), and two sequentially coupled Fabry-Perot interferometers that analyze the frequency shift of the broad-band molecular return signal by means of the so-called double-edge technique (Chanin et al., 1989; Flesia and Korb, 1999; Gentry et al., 2000).





Both, the Rayleigh and Mie channel sample the backscatter signal time-resolved leading to 24 bins with a vertical resolution that can vary between $0.25\,\mathrm{km}$ and $2.0\,\mathrm{km}$ (see Fig. 2). Depending on the number of averaged measurements, which have a horizontal resolution of about $3\,\mathrm{km}$, the horizontal resolution of the wind observations is usually $90\,\mathrm{km}$ for the Rayleigh channel (Rayleigh-clear winds) and $10\,\mathrm{km}$ for the Mie channel (Mie-cloudy winds). Furthermore, due to the high-spectral resolution

receiver configuration, information on the vertical distribution of aerosol and cloud optical properties such as backscatter and extinction coefficients can be retrieved (Ansmann et al., 2007; Flamant et al., 2008; Flament et al., 2021; Feofilov et al., 2022). Further information about the Aeolus satellite, the ALADIN instrument and the retrieval algorithms can be found in (e.g., ESA, 1999; Reitebuch, 2012; Reitebuch et al., 2020; Kanitz et al., 2019; Straume et al., 2018, 2020).

The data of Aeolus is provided in different product levels containing different types of information (e.g., Tan et al., 2008;
ESA, 2016; Tan et al., 2017; Rennie, 2018). The Level 0 data contains the raw data of ALADIN as well as the instrument housekeeping data and the housekeeping data of the satellite platform. The Level 1B (L1B) data provides processed ground echo data and preliminary horizontal LOS (HLOS) wind observations that have not been corrected for atmospheric temperature and pressure (Reitebuch et al., 2018). The L2B data contains the time series of fully processed profiles of HLOS winds along the satellite orbit. L2B Rayleigh-wind data is corrected for atmospheric temperature $T$ and pressure $p$ which is needed to avoid
systematic errors (Dabas et al., 2008). As the Rayleigh-Brillouin spectrum of the molecular scattered light depends on $T$ and $p$ (Witschas et al., 2010; Witschas, 2011a, b; Witschas et al., 2014), any differences between instrument response calibration and wind observation have to be taken into account. L2B data is used by the ECMWF for NWP (Tan et al., 2017; Rennie, 2018) and for the validation by means of 2-$\mu$m DWL and Falcon in-situ wind observations, as presented in Sect. 5. It is worth mentioning that the sign of the HLOS winds is defined such that it is positive for winds blowing away from the satellite
LOS. For instance, for an ascending orbit, when the satellite moves from south to north and the laser is pointing east-wards, westerly winds leadto positive HLOS winds. Furthermore, the L2B winds are classified by means of the optical properties of the atmosphere namely into Rayleigh-clear winds, indicating wind observations in aerosol-poor atmosphere and Mie-cloudy winds, indicating winds acquired from particulate backscatter, predominately from clouds or ground returns. There are also Rayleigh-cloudy and Mie-clear winds available in the data product which are not further discussed within this study.

Both the L1B and L2B processors are continuously updated, modified and improved. Thus, data processed with different processor versions may result in a different HLOS winds. In this study, the second reprocessed data set (processor baseline 11 - L2B processor version L2bP 3.40), is used for the AVATAR-I time frame. For AVATAR-T, the near-real-time (NRT) data which is used in this study was processed with processor baseline 12 (L2bP 3.50). As only minor modifications have been applied between baseline 11 and 12, the different processor versions are not expected to have significant impact on the results from the
two different campaign data sets.

### 3.2 The airborne 2-$\mu$m DWL

The 2-$\mu$m DWL has been operated by DLR for more than 20 years and has been deployed in several ground and airborne field campaigns for measuring for instance aircraft wake vortices (Köpp et al., 2004), aerosol optical properties (Chouza et al., 2015, 2017), horizontal wind speeds over the Atlantic Ocean as input data for assimilation experiments (Weissmann et al.,





2005; Schäfler et al., 2018) and horizontal as well as vertical wind speeds to study the life cycle of gravity waves (Witschas et al., 2017; Witschas et al.). In addition, the system was applied in several Aeolus pre-launch campaigns conducted within the last 10 years (e.g., Marksteiner et al., 2018; Lux et al., 2018).

The 2-$\mu$m DWL is a heterodyne detection wind lidar system based on a Tm:LuAG laser operating at a wavelength of 2022.54 nm (vacuum), a laser pulse energy of 1 mJ to 2 mJ and a pulse repetition rate of 500 Hz, ensuring eye-safe operation.

The system composed of three main units, namely (1) a transceiver head containing the laser, an 11 cm afocal telescope, receiver optics, detectors and a double wedge scanner enabling to steer the laser beam to any position within a 30° cone angle; (2) a power supply and the cooling unit of the laser, mounted in a separate rack; and (3) a rack containing the data acquisition unit and the control electronics. For a more detailed description of the 2-$\mu$m DWL including a listing of the system specifications refer Witschas et al. (2017).

To measure the three-dimensional wind speed and direction, the velocity-azimuth display (VAD) scan technique is applied (Browning and Wexler, 1968). That is, a conical step-and-stare scan around the vertical axes with an off-nadir angle of 20° is performed for 21 LOS positions, separated by 18° in azimuth direction. Considering a 1 s averaging time for each LOS measurement and an additional second in order to change the laser beam pointing direction, one scanner revolution takes about 42 s. By further taking into account the aircraft cruise speed of about 200 ms$^{-1}$, the horizontal resolution of 2-$\mu$m DWL wind

observations is about 8.4 km, depending on the actual ground speed of the aircraft. The vertical resolution of the wind observations is determined by the laser pulse length and the averaging interval which is set to be 100 m.

To retrieve wind speed and wind direction profiles from the single LOS measurements performed during one scanner revolution, an algorithm based on a maximum function of accumulated spectra (MFAS) is used as baseline for the 2-$\mu$m DWL(Witschas et al., 2017). When using the MFAS algorithm, wind speed and wind direction are retrieved without estimating single LOS

wind velocities and thus yields valid wind estimates even in regions with low signal-to-noise ratio (SNR). In particular, the spectra of all 21 LOS measurements are shifted to be proportional to their azimuth angle and an assumed wind vector, and accumulated afterwards. For a correctly assumed wind vector, the accumulated spectra have a maximum and thus indicate the prevailing wind vector. By applying the MFAS algorithm to one scanner revolution, the horizontal and vertical resolution of the retrieved wind vectors is about 8.4 km and 100 m, respectively. To additionally increase the coverage of 2-$\mu$m DWL mea-

surements, the number of accumulated LOS measurements can be further increased, at the expense of lower horizontal and/or vertical resolution.

The suitability of the 2-$\mu$m DWL as a reference instrument for the Aeolus validation was demonstrated by means of dropsonde comparisons during several campaigns in the past. Based on these measurements, it was shown that single 2-$\mu$m DWL LOS measurements have a systematic error below 0.1 ms$^{-1}$ and a random error of about 0.2 ms$^{-1}$. The systematic error of hori-

zontal wind speed is determined to be below 0.1 ms$^{-1}$ and the corresponding random error is about 1 ms$^{-1}$ (Witschas et al., 2020; Weissmann et al., 2005; Chouza et al., 2016; Reitebuch et al., 2017; Schäfler et al., 2018; Witschas et al., 2017).



### 3.3 Falcon nose-boom

In addition to the 2-$\mu$m DWL observations, horizontal and vertical wind speed were measured in-situ at flight level by the Falcon nose-boom which hosts a Rosemount model 858 flow angle sensor that is used together with a Honeywell Lasernav

YG 1779 inertial reference system (Bögel and Baumann, 1991; Krautstrunk and Giez, 2012). Falcon nose-boom observations provide a temporal resolution of $1\,\mathrm{Hz}$ which corresponds to a horizontal resolution of about $200\,\mathrm{m}$, considering the usual cruise-speed of the Falcon aircraft of about $200\,\mathrm{m\,s^{-1}}$. The random error of the horizontal wind speed is specified to be $0.9\,\mathrm{m\,s^{-1}}$, and thus, provides reference data at flight level that is suitable for Aeolus validation. Further details about the calibration method and retrieval algorithms of Falcon in-situ winds are given by Mallaun et al. (2015) and Giez et al. (2017).

## 4  Methodology

### 4.1  Adaption of the measurement grid

Due to the different horizontal and vertical sampling and resolution of 2-$\mu$m DWL measurements ($\approx 8.4$ km, 100 m for one scanner revolution) and Aeolus measurements ($\approx 90$ km (Rayleigh) and $\approx 10$ km (Mie), 0.25 km to 2 km), special averaging procedures are needed to compare respective wind observations that are described by Witschas et al. (2020). Furthermore, as

Aeolus is only providing HLOS winds, the 2-$\mu$m DWL measurements have to be projected onto the Aeolus HLOS direction. To this end, the wind speed and wind direction measured by the 2-$\mu$m DWL are averaged to the Aeolus grid by using the top and bottom altitudes as well as the start and stop latitudes given in the Aeolus L2B data product. As the 2-$\mu$m DWL does not provide full data coverage, a threshold for the number of available 2-$\mu$m DWL observations within an Aeolus grid cell has to be set. In this study, valid 2-$\mu$m DWL measurements need to be available to cover at least 50% of the Aeolus bin to consider the

averaged wind speed and wind direction for further comparison. Other, more restrictive thresholds, for instance 75% or 90%, yields comparable systematic and random errors but with a significantly reduced number of data points that can be compared. For the Falcon in-situ measurements no such a threshold is necessary, because they have full horizontal coverage.

Afterwards, all valid averaged wind speeds ($\mathrm{ws}_{2\mu m}$) and directions ($\mathrm{wd}_{2\mu m}$) are projected onto the horizontal LOS of Aeolus ($v_{2\mu\mathrm{m}_{\mathrm{HLOS}}}$) by means of the range-dependent azimuth angle $\varphi_{\mathrm{Aeolus}}$ that is provided in the Aeolus L2B data product

according to

$$v_{2\mu\mathrm{m}_{\mathrm{HLOS}}} = \cos\left(\varphi_{\mathrm{Aeolus}} - \mathrm{wd}_{2\mu m}\right) \cdot \mathrm{ws}_{2\mu m}. \tag{1}$$

In a next step, the Aeolus HLOS winds (Rayleigh-clear and Mie-cloudy) are extracted for areas of valid 2-$\mu$m DWL measurements. Beforehand, the data is filtered by means of an estimated error (EE) for the wind speeds given in the L2B data product and which is estimated based on the measured signal levels as well as the temperature and pressure sensitivity of

the Rayleigh channel response (Tan et al., 2008; Tan et al., 2017). In this study, EE thresholds of $7.0\,\mathrm{m\,s^{-1}}$ (Rayleigh-clear) and $5.5\,\mathrm{m\,s^{-1}}$ (Mie-cloudy) are used for the AVATAR-I dataset and $8.5\,\mathrm{m\,s^{-1}}$ (Rayleigh-clear) and $5.0\,\mathrm{m\,s^{-1}}$ (Mie-cloudy)



are used for the AVATAR-T dataset. A discussion about the selection of EE thresholds is given in Sect.4.3. The Falcon wind measurements are treated in a similar way.

The explained averaging procedure is illustrated in Fig. 3 by means of the satellite underflight performed on 16 Sept 2019
during the AVATAR-I campaign covering a flight distace of about 1000 km. The top panels show the measured 2-$\mu$m DWL wind speed (a) and direction (b). The corresponding projection onto the Aeolus HLOS direction using Eq. (1) is plotted in panel (c) and the actual Aeolus L2B Rayleigh-clear winds are indicated in panel (d). From the valid 6422 data points acquired by the 2-$\mu$m DWL along the underflight leg with its original resolution, only 163 data points remain for Rayleigh-clear wind comparison and 53 for Mie-cloudy (not shown) after being projected to the Aeolus grid. Thus, multiple underflights are needed to get enough data points for a statistically significant comparison.

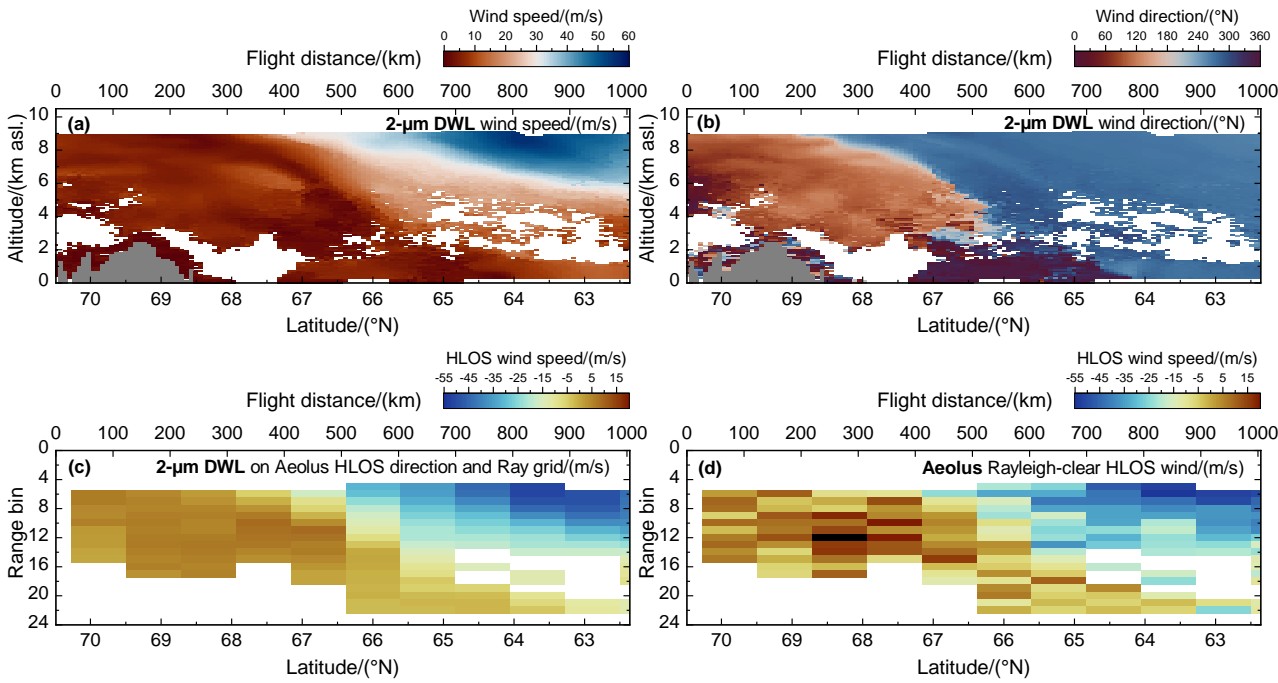

**Figure 3.** 2-$\mu$m DWL wind speed (a) and wind direction (b) for the AVATAR-I underflight on 16 Sept 2019. The corresponding 2-$\mu$m DWL data projection onto the Aeolus HLOS direction is shown in panel (c), and the Aeolus HLOS Rayleigh winds are shown in panel (d). White fields indicate missing data.


## 4.2   Statistical comparison of Aeolus and reference data

To validate the quality of Aeolus HLOS observations ($\mathcal{O}$), the difference to the corresponding reference or background data ($\mathcal{B}$) from the 2-$\mu$m DWL or the Falcon nose-boom, projected onto the Aeolus viewing direction, is calculated according to

$$v_{\text{diff}} = \mathcal{O}_{\text{HLOS}} - \mathcal{B}_{\text{HLOS}}. \tag{2}$$





The bias and standard deviation (STD) of $v_{\text{diff}}$ are calculated by use of

$$\text{bias}_{v_{\text{diff}}} = \frac{1}{n} \sum_{i=1}^{n} v_{\text{diff}} \tag{3}$$

and

$$\text{STD}_{v_{\text{diff}}} = \sqrt{\frac{1}{n-1} \sum_{i=1}^{n} \left(v_{\text{diff}} - \text{bias}_{v_{\text{diff}}}\right)^2}. \tag{4}$$

where $n$ is the number of available data points. In addition to the standard deviation, the scaled median absolute devia-
tion (scaled MAD) is calculated according to

$$\text{scaledMAD}_{v_{\text{diff}}} = 1.4826 \times \text{median}\left(\left|v_{\text{diff}} - \text{median}\left(v_{\text{diff}}\right)\right|\right). \tag{5}$$

The scaled MAD has the advantage that it is less sensitive to single outliers which may result in larger STD values. It is thus
used as a measure of the random error of Aeolus HLOS winds. The scaled MAD is identical to the standard deviation (Eq. (4))
in case the analyzed data is normally distributed.

Furthermore, the uncertainty of the bias $\sigma_{\text{bias}_{v_{\text{diff}}}}$ is calculated according to

$$\sigma_{\text{bias}_{v_{\text{diff}}}} = \frac{\text{scaled MAD}}{\sqrt{n}}. \tag{6}$$

### 4.3 Quality control of Aeolus data

Before performing a statistical comparison, an adequate quality control (QC) of Aeolus data is mandatory. The first parameter
that is used for that purpose is the validity flag in the L2B wind product. Only winds with a validity flag that equals one are
considered for further comparison. An additional parameter for QC is the estimated error (EE) which is reported in the L2B
product. According to the Aeolus L2B Algorithm Theoretical Basis Document (Rennie et al., 2020), the EE for the Rayleigh
HLOS winds is computed from the uncertainty in the Rayleigh spectrometer response and is currently only depending on
the signal level (Poisson noise) and the solar background. Other quanteties as the dependency on atmospheric temperature and
pressure, the contamination by Mie scattering or the detector read-out noise are currently not considered for the EE calculation.
On the other hand, the EE for Mie cloudy winds is derived from the precision of the Mie response which itself is depending on
the accuracy of the applied fit algorithm (Rennie et al., 2020).

As the applied EE threshold impacts the determined statistical parameters such as the systematic and random error, a proper
choice of the EE threshold is crucial. And as the EE is varying over time and geographical location due to the different solar
background and signal levels, the determination of a proper EE threshold gets even more difficult. Ideally, the EE thresholds
are chosen such that the Aeolus wind errors with respect to the validation instrument (i.e. the 2-$\mu$m DWL or the Falcon in-situ





winds) are normally distributed, since the random error is defined as the standard deviation of a Gaussian distribution in the Aeolus mission requirements (ESA, 2016). While the comprhensive approach to detemine the the EE threshold is described separately by Lux et al. (2022b), only a rough outline of the procedure is presented here.

One fundamental aspect when defining an EE threshold is to discard observations that suspiciously deviate from the expectations (outliers). To screen a data set for outliers, it is common to use the so-called Z-score which describes the distance from the mean (bias) in units of the standard deviation according to

$$Z_i = \frac{v_{\text{diff}_{\text{HLOS}}\,i} - \text{bias}(v_{\text{diff}_{\text{HLOS}}})}{\text{STD}_{v_{\text{diff}_{\text{HLOS}}}}}. \tag{7}$$

However, as for instance shown by Iglewicz and Hoaglin (1993), Z-scores are not satisfactory, especially for small data sets as they are available from airborne campaigns. The problem when using the Z-score is, that the mean and standard deviation used for Z-score calculation can be greatly affected by single outliers. To solve this problem, it is useful to apply the modified Z-score instead, which is defined as

$$Z_{m,i} = \frac{v_{\text{diff}_{\text{HLOS}}\,i} - \text{median}(v_{\text{diff}_{\text{HLOS}}})}{\text{scaledMAD}_{v_{\text{diff}_{\text{HLOS}}}}}. \tag{8}$$

Hence, compared to the Z-score, the mean is replaced by the median and the standard deviation is replaced by the scaled MAD, making the modified Z-score more robust with respect to outliers and to the samle size. In this study, a modified Z-score threshold of $Z_m > 3$ is used to define outliers.

For the definition of a suitable EE threshold, it turned out that it is useful to perform a statistical analysis of systematic and random errors and the data coverage depending on the EE threshold. Fig. 4 illustrates the results from the statistical comparison of Aeolus L2B Mie-cloudy (a,c) and Rayleigh-clear winds (b,d) against 2-$\mu$m DWL data (one scan accumulation) from the 10 underflights of the AVATAR-I campaign (top) and the 11 underflights of the AVATAR-T campaign (bottom) in dependence of the EE threshold without and with outlier removal based on the modified Z-score. The bar plots depict the percentage of filtered winds after QC ($Z_m < 3$, green and blue bars) and outliers ($Z_m > 3$, red bars) from all wind results that are flagged valid in the L2B product (left y-axes). The percentage of outliers is indicated above the bars. The lines and symbols refer to the statistical results (mean bias [Eq (3)], standard deviation [Eq (4)] and scaled MAD [Eq (5)])) without removing the gross errors from the datasets (gray, light-blue, magenta), while the black, blue and red lines represent the statistical parameters after QC based on the modified Z-score (right y-axes). The EE thresholds that are deemed reasonable to provide robust statistical results are highlighted by orange frames. As expected, the number of available valid data points increases when increasing the EE threshold. This is true for both, Rayleigh-clear (green bars) and Mie cloudy winds (blue bars) and for both campaign data sets. At the same time, also the number of outliers in the data set increases. Additionally, it can be seen that the mean bias (black without and gray with outliers), the standard deviation (dark-blue without and light-blue with outliers) and the scaled MAD (red without and magenta with outliers) are depending on the EE threshold. This further demonstrates that the EE threshold can significantly impact the results of the statistical comparison.





**Figure 4.** Results from the statistical comparison of Aeolus L2B Mie-cloudy (a,c) and Rayleigh-clear winds (b,d) against 2-$\mu$m DWL data (one scan accumulation) are shown from the 10 underflights of the AVATAR-I campaign (top) and the 11 underflights of the AVATAR-T campaign (bottom), depending on the EE threshold without and with outlier removal based on the modified Z-score. The bar plots depict the portion of filtered winds after QC ($Z_m < 3$, green and blue bars) and gross errors ($Z_m > 3$, red bars) from all wind results that are flagged valid in the L2B product (left y-axes). The percentage of gross errors is indicated above the bars. The lines and symbols refer to the statistical results (mean bias, standard deviation and scaled MAD) without removing the gross errors from the datasets (gray, light-blue, magenta), while the black, blue and red lines represent the statistical parameters after QC based on the modified Z-score (right y-axes). The EE thresholds that are deemed reasonable to provide robust statistical results are highlighted by orange frames.

In the following, several subjectively selected quality criteria are used to define a suitable EE threshold. For instance, it is checked for which EE threshold the STD starts to deviate from the scaled MAD as this marks the point where the data set starts





to deviate from a normal distribution. Furthermore, the number of available data points and determined outliers is analyzed. If
too many outliers are determined, the QC can be considered too strict. Additionally, it is checked if the respective statistical
quantities differ significantly when being calculated from the data set with and without outliers.

For the Rayleigh-clear winds of the AVATAR-I data set (Fig. 4 b) for instance, the number of available data points increases
quickly with increasing EE threshold. For an EE threshold $4.5\,\mathrm{m\,s^{-1}}$, already 80% of all data points are included and only 0.4%
outlier are detected. Furthermore, for this threshold, the STD and the scaled MAD are almost similar for both cases calculated
with and without outliers. Between an EE threshold of $6.0\,\mathrm{m\,s^{-1}}$ and $6.5\,\mathrm{m\,s^{-1}}$, the STD calculated including outliers makes a
jump of about $1\,\mathrm{m\,s^{-1}}$, whereas the other quantities remain rather constant. For an EE threshold larger than $7.0\,\mathrm{m\,s^{-1}}$, mainly
outliers are added to the data set. Thus, $7.0\,\mathrm{m\,s^{-1}}$ is considered as the optimum EE threshold for Rayleigh-clear winds of the
AVATAR-I data set.

For the Rayleigh-clear winds of the AVATAR-T data set, which was acquired almost two years later in a different geograph-
ical region and at decreased ALADIN signal performance, the distribution looks different (Fig. 4 d). The number of valid data
points increases much slower compared to the AVATAR-I case. For an EE threshold of $7.0\,\mathrm{m\,s^{-1}}$, the STD is still rather close
to the scaled MAD, however, only 65% of the data points are included. By further increasing the EE threshold to $8.5\,\mathrm{m\,s^{-1}}$,
the number of used data points increases to 78%. For even larger EE thresholds, the scaled MAD calculated without (red)
and with (magenta) outliers starts to increase, indicating that the outliers start to have an impact on the calculated statistical
parameters. Thus, an EE threshold of $8.5\,\mathrm{m\,s^{-1}}$ seems to be a good compromise for the Rayleigh-clear winds of the AVATAR-T
data set.

For the Mie-cloudy data set, the distribution of statistical parameters is even more sensitive to the EE threshold. It can be
seen that Mie-cloudy winds in general contain more outliers. This also confirms that the QC by means of the validity flag and
the EE threshold is not sufficient and that an additional QC by means of the modified Z-score is needed, especially for the
Mie-cloudy winds (Lux et al., 2022b). Following the same logic as for the Rayleigh-clear winds, an optimal EE threshold of
$5.5\,\mathrm{m\,s^{-1}}$ is determined for the Mie-cloudy winds of the AVATAR-I data set, and $5.0\,\mathrm{m\,s^{-1}}$ for the Mie-cloudy winds of the
AVATAR-T data set.

## 5 Statistical comparison

A scatter plot of Aeolus HLOS wind speeds versus 2-$\mu$m DWL data is shown in Fig. 5 for the AVATAR-I data set (left) as well
as for the AVATAR-T data set (right). Rayleigh-clear winds and Mie-cloudy winds are indicated by blue dots and orange dots,
respectively and corresponding line fits are depicted by the light blue and yellow lines. The $x = y$ line is represented by the
gray dashed line. Altogether, the 10 underflights during the AVATAR-I campaign provide 1155 valid data points for Rayleigh-
clear wind validation and 701 valid data points for Mie-cloudy wind validation. The QC identified 18 (1.6%) data points of
the Rayleigh-clear data set and 30 (4.3%) data points of the Mie-cloudy data set as outliers as they exceeded the modified
Z-score threshold of 3 according to Eq. (8). The 11 underflights during AVATAR-T resulted in 465 and 144 data points for
Rayleigh-clear and Mie-cloudy wind validation, respectively, where the modified Z-score threshold lead to the identification



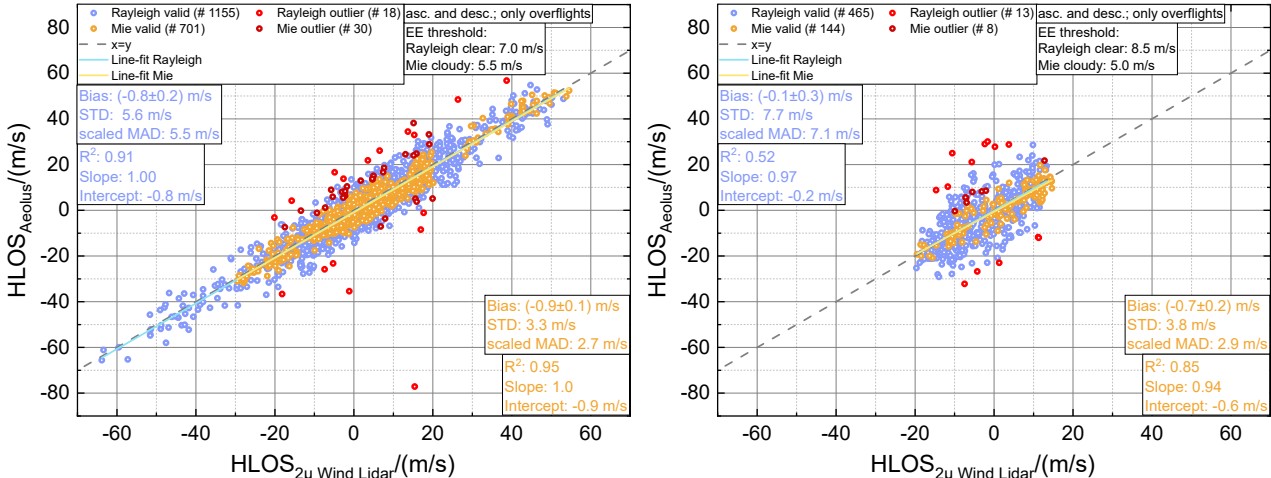

**Figure 5.** Aeolus HLOS wind speed plotted against the 2-$\mu$m DWL wind speed projected onto the horizontal viewing direction of Aeolus for the 10 underflights performed during the AVATAR-I campaign (left) and for the 11 underflights performed during the AVATAR-T campaign (right). The wind measurements are separated in Rayleigh-clear winds (blue) and Mie-cloudy winds (orange). Outliers that exceeded a modified Z-score threshold of 3 are indicated by light-red and dark-red points, respectively. Corresponding least-square line fits are indicated by the light blue and yellow line, respectively. The fit results are shown in the insets. The $x = y$-line is represented by the gray-dashed line.

of 13 (2.8%) and 8 (5.6%) outliers. The decreased number of data points observed during the last six AVATAR-T underflights was due to the fact that the 2-$\mu$m DWL was degrading during the campaign period. The degradation itself was caused by the large temperature and humidity gradients causing the transceiver unit to get misaligned. Furthermore, the Aeolus RBS (see also Fig.2), with thicker but fewer range gates in the troposphere for the sake of increased SNR, was less favorable for a good

grid overlap with airborne data which is only acquired from flight altitudes of about 10 km above sea level down to the ground. Additionally, the overall signal levels during the AVATAR-T period were only about half of the one acquired during AVATAR-I, caused by a degrading Aeolus instrument performance.

## 5.1 Systematic error

The mean systematic error of the Aeolus wind data and its corresponding uncertainty are calculated according to Eq. (3) and Eq. (6), respectively. It yields values of $(-0.8 \pm 0.2)\,\mathrm{m\,s^{-1}}$ (Rayleigh-clear) and $(-0.9 \pm 0.1)\,\mathrm{m\,s^{-1}}$ (Mie-cloudy) for the AVATAR-I data set and $(-0.1 \pm 0.3)\,\mathrm{m\,s^{-1}}$ (Rayleigh-clear) and $(-0.7 \pm 0.2)\,\mathrm{m\,s^{-1}}$ (Mie-cloudy) for the AVATAR-T data set, respectively. Hence, the systematic errors for both wind products and both campaign periods are close to the specified mission requirement of $0.7\,\mathrm{m\,s^{-1}}$ for Aeolus HLOS winds (ESA, 2016). Compared to the previous campaign results where the

systematic error was determined to be $(2.1 \pm 0.3)\,\mathrm{m\,s^{-1}}$ (Rayleigh-clear) and $(2.3 \pm 0.2)\,\mathrm{m\,s^{-1}}$ (Mie-cloudy) for the WindVal III data set and $(-4.6 \pm 0.2)\,\mathrm{m\,s^{-1}}$ (Rayleigh-clear) and $(-0.2 \pm 0.1)\,\mathrm{m\,s^{-1}}$ (Mie-cloudy) for AVATAR-E, a significant decrease of the systematic error can be observed (see also Table 3) which is due to the implementation of correction schemes for the hot pixels and the thermal fluctuations on the telescope mirror in the Aeolus processor that were not available in the early





phase of the mission. It is worth mentioning that for the analysis of the AVATAR-I campaign, which was performed in fall
2019, the second reprocessed Aeolus data set is used (B11), and hence, also containing the correction scheme for the telescope
temperature fluctuations. The Aeolus processor versions used for the analysis of the respective campaign data sets are also
given in Table 3. It is worth mentioning that derived values are depending on the applied QC procedure and outlier selection
and thus not necessariely comparable for different campaign data sets.

## 5.2 Random error

The random error of the Aeolus wind is represented by the scaled MAD according to Eq. (5). It is determined to be $5.5\,\mathrm{m\,s^{-1}}$
(Rayleigh-clear) and $2.7\,\mathrm{m\,s^{-1}}$ (Mie-cloudy) for the AVATAR-I data set and $7.1\,\mathrm{m\,s^{-1}}$ (Rayleigh-clear) and $2.9\,\mathrm{m\,s^{-1}}$ (Mie-
cloudy) for the AVATAR-T. The impact of the 2-$\mu$m DWL uncertainty of about $1\,\mathrm{m\,s^{-1}}$ on the determined random error is only
marginal. It can be recognized that the mean random error of Rayleigh-clear winds is significantly larger than the $2.5\,\mathrm{m\,s^{-1}}$
originally specified for Aeolus HLOS winds in altitudes between 2 km and 16 km (ESA, 2016; Kanitz et al., 2019; Reitebuch
et al., 2020). The main reason for this is the lower signal levels of the backscattered light from the atmosphere which is
investigated to be caused by a combination of instrumental misalignment, the wavefront error of the $1.5$ m telescope and laser
induced contamination (LIC) within the system.

Furthermore, it can be seen that the Rayleigh-clear random error increased by about 30% between the AVATAR-I and
the AVATAR-T campaign, although the actual laser UV energy was 20% larger during the AVATAR-T campaign. However,
the atmospheric signal level itself was about 50% smaller due to the aforementioned degradation. This signal decrease was
partly compensated by enlarging the Aeolus range bins during the AVATAR-T campaign which were 750 m instead of 500 m
applied during AVATAR-I. Additionally, the solar background signal was about a factor of three smaller during AVATAR-T
which also partly compensates the overall signal decrease. The mean useful signal, which denotes the average signal level per
observation in LSB (least significant bit) after being corrected for the detection chain offset (DCO), the solar background and
the dark current is determined to be $(247\pm1)\,\mathrm{LSB}$ for Rayleigh signals during AVATAR-I and $(175\pm3)\,\mathrm{LSB}$ for AVATAR-T.
Considering just Poisson noise in the measured data set, the random error can be considered to be proportional to $N^{-1/2}$,
where $N$ is the signal level. Hence, by using the random error determined from AVATAR-I and considering the mean useful
signal levels of both campaign data sets, the random error for AVATAR-T would be expected to be $(247/175)^{1/2}\times5.5\,\mathrm{m\,s^{-1}} =$
$6.5\,\mathrm{m\,s^{-1}}$, which is at least close to the measured value of $7.1\,\mathrm{m\,s^{-1}}$. It should also be mentioned that the error calculation
based on signal levels is only a rough approximation as valid Rayleigh-clear winds are also available in aerosol-loaded areas
with a scattering ratio larger than 2 or 3. Hence, the signal is not necessarily origination from backscattering on molecules
which will distort the random error calculation of the Rayleigh-clear winds.

## 5.3 Falcon in-situ measurements

In addition to 2-$\mu$m DWL observations, in-situ measurements performed on-board the Falcon aircraft were statistically ana-
lyzed as demonstrated by the scatter plot shown in Fig. 6. Here, the Aeolus HLOS Rayleigh-clear wind speeds are plotted
versus Falcon in-situ data for the AVATAR-I data set (blue dots) as well as for the AVATAR-T data set (green dots). Line fits





**Table 3.** Aeolus systematic and random error determined from different campaign data sets.

|  | Rayleigh-clear | | | Mie-cloudy | | | |
|  | Bias (m/s) | sc. MAD (m/s) | points | Bias (m/s) | sc. MAD (m/s) | points | Processor version |
|---|---|---|---|---|---|---|---|
| WindVal III* | $2.1 \pm 0.3$ | 4.0 | 231 | $2.3 \pm 0.2$ | 2.2 | 109 | L2bP 3.01 (B02) |
| AVATAR-E* | $-4.6 \pm 0.2$ | 4.4 | 504 | $-0.2 \pm 0.1$ | 2.2 | 339 | L2bP 3.10 (B03) |
| AVATAR-I | $-0.8 \pm 0.2$ | 5.5 | 1155 | $-0.9 \pm 0.1$ | 2.7 | 701 | L2bP 3.40 (B11) |
| AVATAR-T | $-0.1 \pm 0.3$ | 7.1 | 465 | $-0.7 \pm 0.2$ | 2.9 | 144 | L2bP 3.50 (B12) |

\* Values taken from (Witschas et al., 2020).

to the data are depicted by the dark blue and and green lines, and the $x = y$ line is represented in gray dashed. The Falcon measurements are mainly taken in altitudes between $10\,\mathrm{km}$ and $11\,\mathrm{km}$ and have a random error of $0.9\,\mathrm{m\,s^{-1}}$ with a temporal resolution of $1\,\mathrm{s}$, hence, providing a very good reference. Solely the representativeness error is difficult to asses and depending
on the vertical homogeneity of the atmosphere in the vicinity of the performed observations. Furthermore, as no Mie-cloudy winds were present at flight level only Rayleigh-clear winds could be analyzed. The mean systematic error is determined to

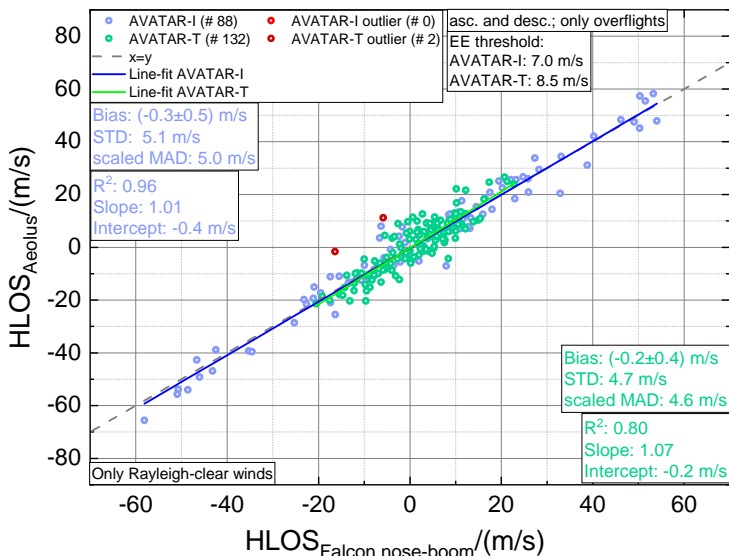

**Figure 6.** Aeolus Rayleigh-clear HLOS winds plotted against the Falcon nose-boom wind speed projected onto the horizontal viewing direction of Aeolus for the 10 underflights performed during the AVATAR-I campaign (blue) and for the 11 underflights performed during the AVATAR-T campaign (green). Outliers that exceeded a modified Z-score threshold of 3 are indicated by light-red and dark red points, respectively. Corresponding least-square line fits are indicated by the dark blue and light green line, respectively. The fit results are shown in the insets. The $x = y$-line is represented by the gray-dashed line.

be $(-0.3 \pm 0.5)\,\mathrm{m\,s^{-1}}$ (AVATAR-I) and $(-0.2 \pm 0.4)\,\mathrm{m\,s^{-1}}$ (AVATAR-T) and hence confirms the results obtained from the





2-$\mu$m DWL analysis and that the Aeolus Rayleigh-clear winds meet the mission requirement of a systematic error smaller than $0.7\,\mathrm{m\,s^{-1}}$.

The corresponding random errors yield values of $5.0\,\mathrm{m\,s^{-1}}$ (AVATAR-I) and $4.6\,\mathrm{m\,s^{-1}}$ (AVATAR-T). Thus, for the AVATAR-I data set, the random error is comparable to the one determined from the 2-$\mu$m DWL ($5.5\,\mathrm{m\,s^{-1}}$), but differs significantly for the AVATAR-T data set, where the 2-$\mu$m DWL analysis yields a random error of $7.1\,\mathrm{m\,s^{-1}}$. This is explained by the fact that the Rayleigh-clear random error depends on the signal level which is lower at lower altituds (see also Sect. 6). For the AVATAR-T campaign, lower signal levels in the Saharan Air Layer (SAL) due to the extinction induced by aerosols cause the random error

of Rayleigh-clear winds to increase in this region. Hence, the mean random error is larger in lower altitudes compared to the one at flight level.

## 6   Aeolus error dependency

In Sec. 5.1 and Sec. 5.2, the mean systematic and random error was determined for the entire data set of the AVATAR-I and the AVATAR-T campaign and for both, Rayleigh-clear and Mie-cloudy winds. In this section, the dependency of these errors

on the actual wind speed, represented by the 2-$\mu$m DWL observations is investigated to verify if the Aeolus calibration routine is working for the entire wind speed range. Moreover, the error dependency on the geographical location is investigated. To study the representativity of the comparison the 2-$\mu$m DWL observations, which may have a temporal distance to the Aeolus overflight of up to 1 hour, the error dependency on the time difference between 2-$\mu$m DWL observation and Aeolus overflight is analyzed. Further, it is verified if the error has any dependency on the scattering ratio, which might be induced by a cross talk

between the signals from the Rayleigh and the Mie channel, respectively. In addition to that, the mean useful signal, the EE, the scattering ratio as well as the actual wind speed are analyzed depending on altitude. These analyses are separated in Rayleigh-clear winds (Sect. 6.1) and Mie-cloudy winds (Sect. 6.2) and provide further insights into the Aeolus error charateristics.

### 6.1   Rayleigh-clear winds

In Fig. 7, the dependency of the Aeolus Rayleigh-clear HLOS wind speed error with respect to the 2-$\mu$m DWL (Eq. (2)) on the

2-$\mu$m DWL measured wind speed (a), on the latitude (b), on the time difference between Aeolus and 2-$\mu$m DWL observation (c) and on scattering ratio (d) is shown. The data from AVATAR-I is indicated in blue and the one of AVATAR-T in orange. The solid lines denote the median value for certain intervals, and the shaded area represents the median $\pm$ the scaled MAD. To have enough data points and hence a reliable value for the mean and the scaled MAD, the averaging intervals $\Delta$ for the AVATAR-I/AVATAR-T data set are chosen to be $\Delta = (5.8/1.7)\,\mathrm{m\,s^{-1}}$ (2-$\mu$m DWL wind speed), $\Delta = (0.44/0.67)°$ (latitude),

$\Delta = (296/342)\,\mathrm{s}$ (time difference) and $\Delta = 0.95/0.98$ (scattering ratio), respectively. From Fig. 7 (a), it is obvious that the Aeolus wind speed error has no significant dependency on the actual wind speed which is represented by the 2-$\mu$m DWL. Both, the median (systematic error) and the scaled MAD (random error) are nearly constant for all wind speeds and both campaign data sets. Please note that the median is used instead of the mean to be insensitive to single outliers of the small number of data for each averaging interval. Still, single outliers for very high wind speeds during the AVATAR-I campaign are an artifact

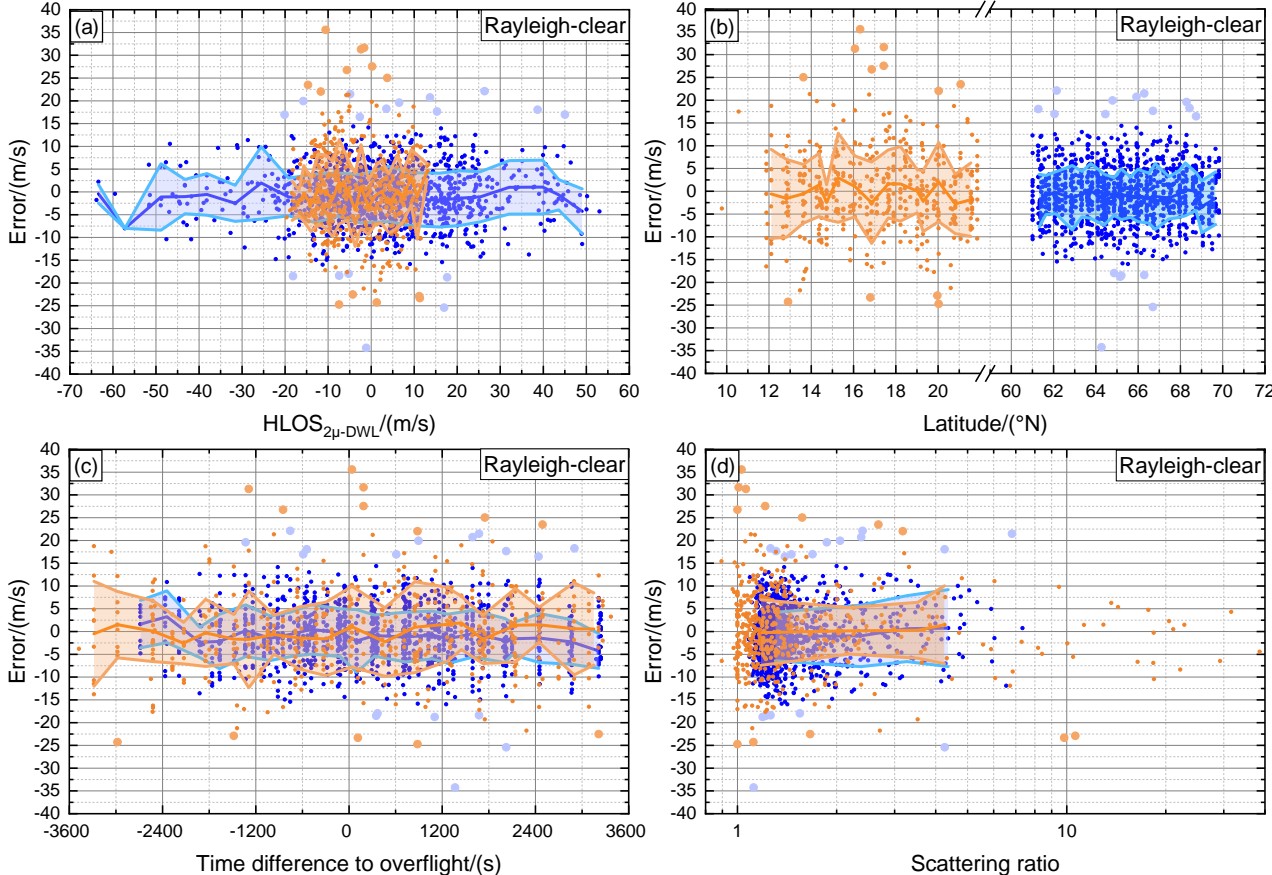

**Figure 7.** Aeolus Rayleigh-clear HLOS wind speed error wrt to 2-$\mu$m DWL measurements depending on 2-$\mu$m DWL measured wind speed (a), on latitude (b), on time difference between Aeolus and 2-$\mu$m DWL observation (c) and on scattering ratio (d). The data set of the AVATAR-I campaign is indicated in blue, the one of AVATAR-T in orange. The solid lines denote the median value for certain intervals, and the shaded area represents the median $\pm$ scaled MAD. Large dots represent outliers which were identified by modified Z-score with a threshold of 3 that is calculated for the respective averaging interval.

caused by a lack of sufficient data points. Furthermore, for the AVATAR-I data set, it is evident that there are more outliers with an positive error than with a negative one. A potential explanation for this behavior is not available so far.

In Fig. 7 (b), the error is plotted against the latitude. As both campaign sites were at different latitudes, the x-axis has a break between 22°N and 59°N, but both sides cover the same range of 13°. As for the wind speed, no significant dependency of the systematic and random error on the latitude can be recognized. Probably the usual range covered during Falcon-campaigns ($\approx$

10°) is not enough to resolve any geolocation dependency of the Aeolus errors after the M1 bias correction became active.

In addition, it is investigated if the time difference between Aeolus overpass and the actual 2-$\mu$m DWL observation on the track has an impact on the determined wind speed error (Fig. 7 (c)). It can be seen that the maximum temporal discrepancy between Aeolus overpass and 2-$\mu$m DWL observation is always smaller than 1 hour, and that neither the systematic nor the random error is significantly depending on this time difference.





The dependency of the Aeolus errors on the scattering ratio is shown in Fig. 7 (d). It is obvious that Rayleigh-clear winds
are even available for backscattering ratios of 10 and larger. For such high values it can not be excluded that a cross-talk
between the Mie channel and the Rayleigh channel leads to an enhanced systematic error. However, such a behavior could not
be confirmed from this analysis. It can be seen that outliers, determined by the modified Z-score threshold of 3, appear for low
as well as for high backscatter ratios. Additionally, the mean systematic and random error show no remarkable dependency on
the backscattering ratio.

   Hence, the Aeolus mean systematic and random errors are neither significantly depending on the wind speed, nor on the
latitude, the time difference and the scattering ratio, confirming that the Aeolus calibration is working properly.

   In addition, the altitude dependency of various quantities was investigated. In Fig. 8, the altitude dependency of the mean
useful signal (a), the estimated error (b), the Aeolus error with respect to the 2-$\mu$m DWL observations (c), the scattering ra-
tio (d), and the HLOS wind velocity derived from 2-$\mu$m DWL measurements (e) for Rayleigh-clear wind observations available
for the AVATAR-I data set (blue) and the AVATAR-E dataset (orange) are shown. The mean useful signal denotes the mean
signal level per observation and range bin in LSB after being corrected for the detection DCO, the solar background and the
dark current. For the AVATAR-I data set, the scattering ratio values are taken from the L2B data. For the AVATAR-T data set,
however, scattering ratio values were partly set to 1 to avoid problems with the assimilation in the ECMWF model. Hence, the
scattering ratio values are calculated from L1B after adaptation to the L2B grid and averaging as it was done for the wind pro-
cessing. The actual valid data points are indicated by the small dots and corresponding outliers, defined by a modified Z-score
threshold of 3, are plotted by larger dots. The median value per each range gate is indicated by the solid line, and the shaded
area indicates the median $\pm$ the scaled MAD for each range bin. By analyzing the altitude dependency of the mean useful
signal (Fig. 8 (a)) it can be seen that the two data sets differ. For AVATAR-I (blue), the signal levels are rather constant at about
$240\,\mathrm{LSB}$. Only at altitudes of about $10.5\,\mathrm{km}$, the signal level is twice as high, due to the larger range bin size at this altitude (see
also Fig. 2). On the contrary, the mean signal level for the AVATAR-T data shows a remarkable decrease between about $4.5\,\mathrm{km}$
and ground, which is due to the signal extinction caused by the aerosols that are prominent in the SAL. Even at altitudes above
$4.5\,\mathrm{km}$, the signal levels for AVATAR-T data were lower than for AVATAR-I, although the range bins were already increased
to $750\,\mathrm{m}$. This additionally confirms that the decreasing Aeolus performance leads to lower (Rayleigh-clear) signal levels at all
altitudes. Furthermore, it can be observed that outliers appear in all altitudes and were successfully determined by the applied
Z-Score threshold.

   The altitude dependency of the EE is plotted in Fig. 8 (b). It can be seen that it is indirectly proportional to the signal levels
as shown in panel (a). All regions of smaller signal levels correspond to a larger EE (as expected). For AVATAR-I, the EE is
about $4\,\mathrm{m\,s^{-1}}$ at all altitudes except for the heighest range bin at about $10.5\,\mathrm{km}$ where is goes down to $2.5\,\mathrm{m\,s^{-1}}$ due to the
larger range bin size. At lower altitudes it goes up to $5\,\mathrm{m\,s^{-1}}$ due to the lower signal levels in this region. For the AVATAR-T
data set, the EE is about $4\,\mathrm{to}\,4.5\,\mathrm{m\,s^{-1}}$ between $4.5\,\mathrm{km}$ and $10.5\,\mathrm{km}$ altitude and increases up to $7.5\,\mathrm{m\,s^{-1}}$ below due the low
signal levels in this region. This also explains why the random error retrieved from the Falcon observations in about $10.5\,\mathrm{km}$
altitude ($5.0\,\mathrm{m\,s^{-1}}$) is significantly lower than the mean random error derived from 2-$\mu$m DWL observations ($7.1\,\mathrm{m\,s^{-1}}$).





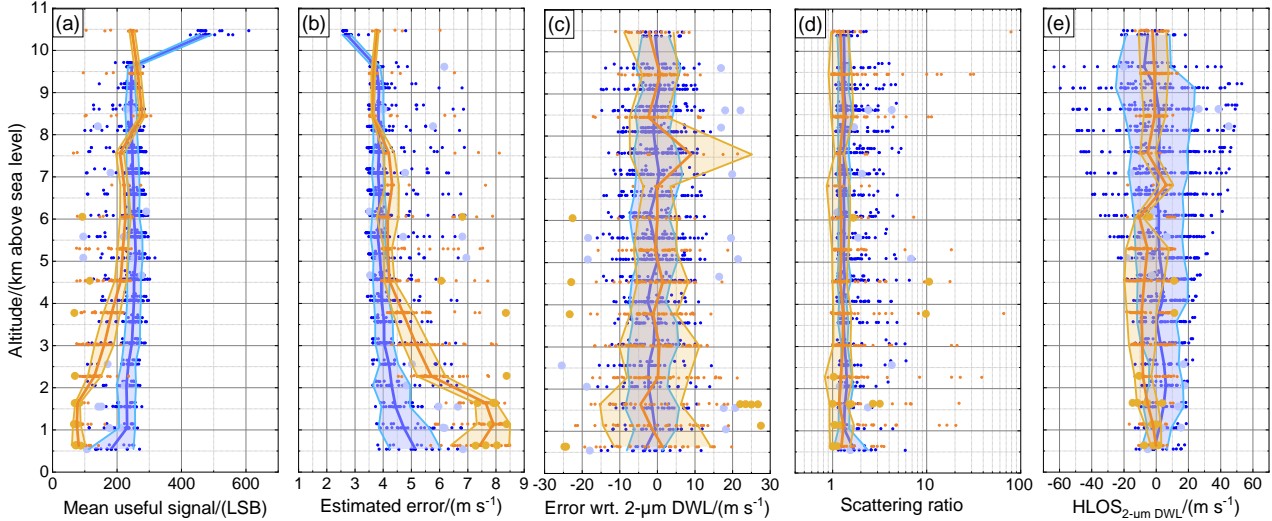

**Figure 8.** Altitude dependency of the mean useful signal (a), the estimated error (b), the Aeolus error wrt to the 2-$\mu$m DWL observations (c), the scattering ratio (d), and the HLOS wind velocity derived from 2-$\mu$m DWL measurements (e) for Rayleigh-clear wind observations available for the AVATAR-I data set (blue) and the AVATAR-E dataset (orange). The actual valid data set is indicated by the small points, the outliers defined by the modified Z-score threshold of 3 are plotted by larger dots. The median value per each range gate is indicated by the solid line, and the shaded area indicates the median $\pm$ scaled MAD for each range gate.

In Fig. 8 (c), the Aeolus error with respect to the 2-$\mu$m DWL is shown. It can be observed that the mean bias does not show any height dependency for both data sets. The outlier at 7.5 km of the AVATAR-T data set is a result of an insufficient number of data points. Furthermore, it can be seen that the random error follows the EE (Fig. 8 (b)) rather well. Especially the random error increase for AVATAR-T in the SAL is well predicted by the EE.

The height dependency of the scattering ratio is depicted in Fig. 8 (d). It can be realized that Rayleigh-clear winds are available even for scattering ratios up 10 and larger, and that these scattering ratios occur in all altitudes. However, the mean scattering ratio is about 1.5 for both data sets and all altitudes.

In Fig. 8 (e) the 2-$\mu$m DWL winds are shown range resolved, demonstrating that much higher wind speeds between $-60$ and $55\,\mathrm{m\,s^{-1}}$ were measured during AVATAR-I (blue) and mainly in altitudes between 6 km and 10 km in the vicinity of the North Atlantic jet stream. During AVATAR-T, the measured wind speeds were much lower varying between $-20$ and $18\,\mathrm{m\,s^{-1}}$ and had their maximum between 3 km and 6 km which is most probably related to the African Easterly Jet.

## 6.2 Mie-cloudy winds

A similar analysis as shown for Rayleigh-clear winds in Fig. 7 is done for Mie-cloudy winds as presented in Fig. 9. Since less data points are available for Mie-cloudy winds, the averaging intervals $\Delta$ for the AVATAR-I/AVATAR-T data set had to be enlarged to $\Delta = (10.5/4.3)\,\mathrm{m\,s^{-1}}$ (2-$\mu$m DWL wind speed), $\Delta = (1.2/1.6)°$ (latitude), $\Delta = (604/677)\,\mathrm{s}$ (time difference) and $\Delta = 6.9/7.4$ (scattering ratio), respectively. The dependency of the Aeolus wind error on the actual wind speed represented

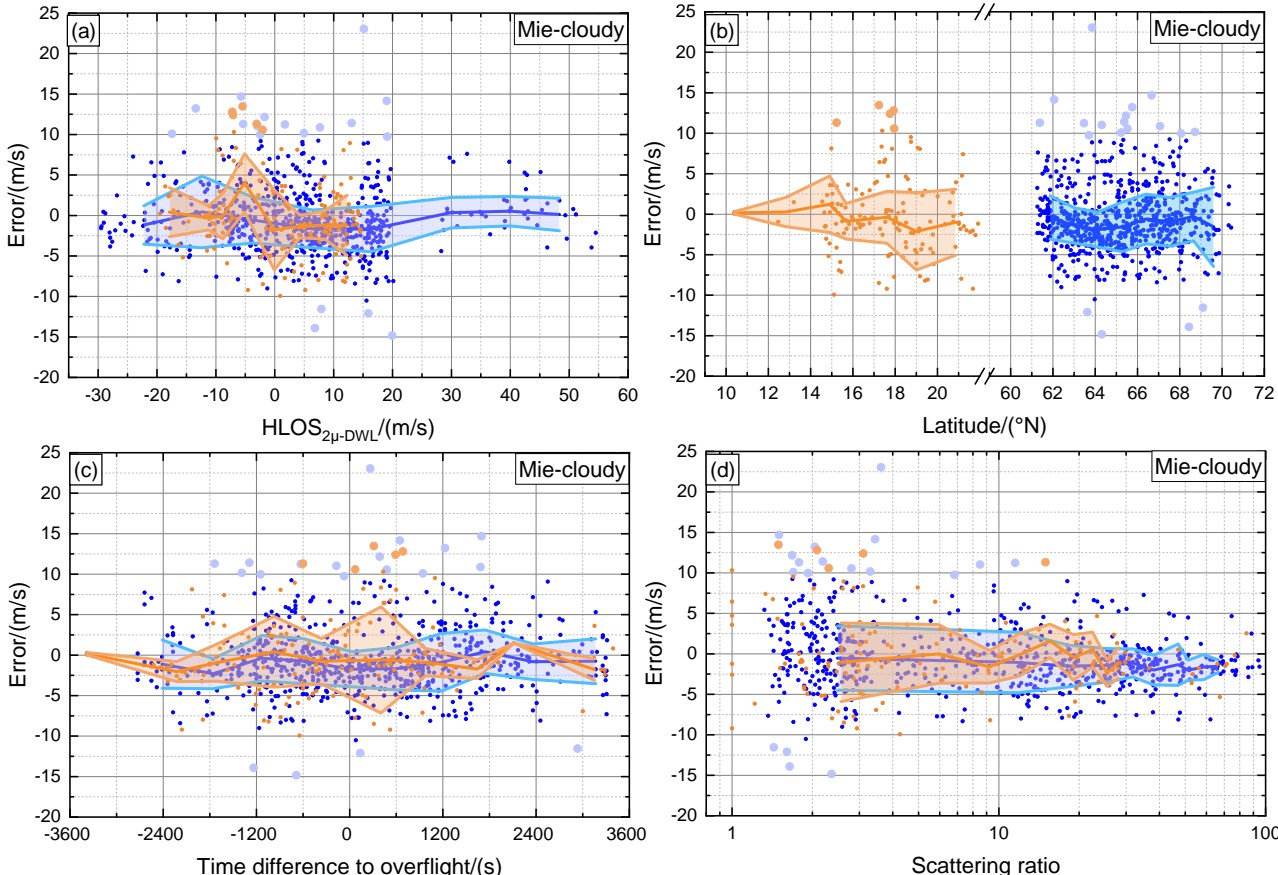

**Figure 9.** Aeolus Mie-cloudy HLOS wind speed error wrt to 2-$\mu$m DWL measurements depending on 2-$\mu$m DWL measured wind speed (a), on Latitude (b), on time difference between Aeolus and 2-$\mu$m DWL observation (c) and on scattering ratio (d). The data set of the AVATAR-I campaign is indicated in blue, the one of AVATAR-T in orange. The solid lines denote the median value for certain intervals, the shaded areas represent the median $\pm$ scaled MAD. Large dots represent outliers which were identified by modified Z-score with a threshold of 3 that is calculated for the respective averaging interval.

by the 2-$\mu$m DWL is shown in Fig. 9 (a). For the AVATAR-I data set it can be recognized that there are regions where the error is negative as for instance for an 2-$\mu$m DWL measured wind speed around $15\,\mathrm{m\,s}^{-1}$, where the systematic error is about $-2\,\mathrm{m\,s}^{-1}$. As this is also the region with the most data points, this may explain the overall negative systematic error of $-0.9\,\mathrm{m\,s}^{-1}$ retrieved from the statistical comparison. Furthermore, it can be seen that there are remarkably more outliers

towards positive errors. This was already true for Rayleigh-clear winds, however, it can not be concluded that this is due to the same root-cause, which is essentially unknown and a topic for further studies.

The dependency of the Aeolus Mie-cloudy wind speed error on latitude is indicated in Fig. 9 (b). For the AVATAR-I data set (blue), it can be seen that the median is negative for all latitudes, varying between $0\,\mathrm{m\,s}^{-1}$ and $-2\,\mathrm{m\,s}^{-1}$. The modulation of the median is not meaningful due to a lack of sufficient data points. This is even more true for the AVATAR-T data set, where

no conclusion can be drawn from the latitude averaged data set.



In Fig. 9 (c), the Aeolus Mie-cloudy error is plotted with respect to the time difference between Aeolus overpass and 2-$\mu$m DWL observation. As for the Rayleigh-clear winds, no significant dependency can be observed for both data sets. Thus, even Mie-cloudy winds seem to change only marginally within a $\pm1$ h time frame (on average), although Mie-cloudy winds are expected to have a higher variability compared to Rayleigh-clear winds.

Fig. 9 (d) depicts the Aeolus error depending on the scattering ratio. From the AVATAR-I data set, it can be seen that the scattering ratio extends to values of up to 100 and that the median drifts to negative values for larger scattering ratios. Also the random error, represented by the scaled MAD, reduces significantly from about $\approx \pm4\ \mathrm{m\,s^{-1}}$ for lower scattering ratios (0-10) to $\approx \pm2\ \mathrm{m\,s^{-1}}$ for larger backscattering ratios (20-100). Hence, it can be concluded that the accuracy and precision of Mie-cloudy winds is depending on the scattering ratio. In particular, observations with a higher backscatter ratio and thus, a better 530   SNR, provide more accurate Mie-cloudy winds. The similar behavior can be seen from the AVATAR-T data set, although it is less conclusive due to the lower number of available data points.

    Furthermore, similar to the Rayleigh-clear wind analysis, the altitude dependency of the mean useful signal (a), the estimated error (b), the Aeolus error wrt to the 2-$\mu$m DWL observations (c), the scattering ratio (d), and the HLOS wind velocity derived 535   from 2-$\mu$m DWL measurements (e) is investigated for Mie-cloudy winds from the AVATAR-I data set (blue) and the AVATAR-T data set (orange), as shown in Fig. 10. From Fig.10 (a) it can be seen that the mean useful signal varies much more than

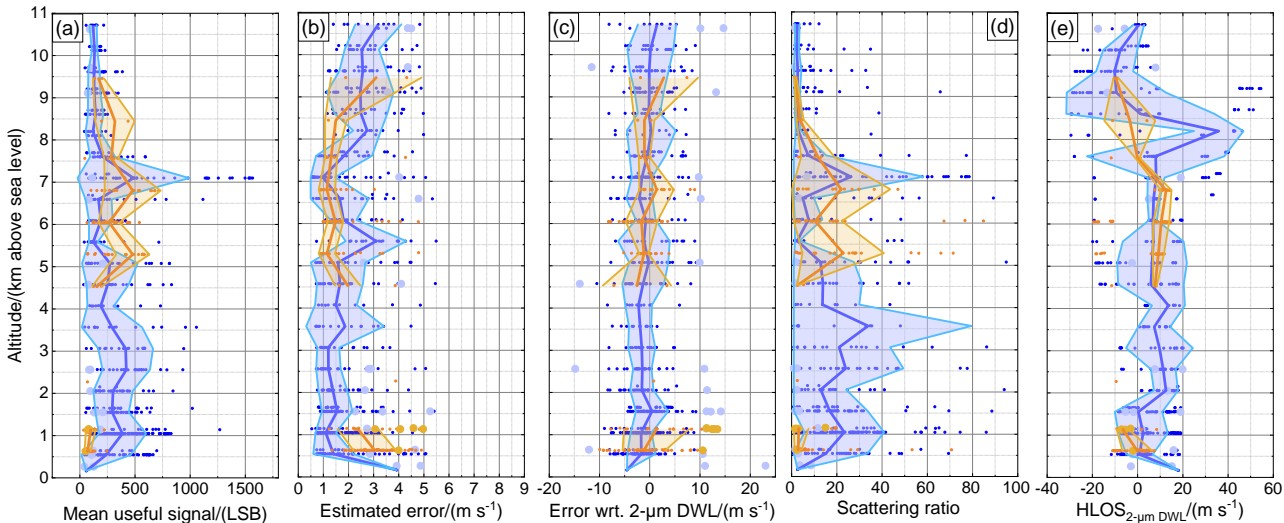

**Figure 10.** Dependency of the mean useful signal (a), the estimated error (b), the Aeolus error wrt to the 2-$\mu$m DWL observations (c), the scattering ratio (d), and the HLOS wind velocity derived from 2-$\mu$m DWL measurements (e) for Mie-cloudy wind observations available for the AVATAR-I data set (blue) and the AVATAR-E dataset (orange). The actual valid data set is indicated by the small points, the outliers defined by the modified Z-score threshold of 3 are plotted by larger dots. The median value per each range gate is indicated by the solid line, and the shaded area indicates the median $\pm$ scaled MAD for each range gate.

for Rayleigh-clear winds from almost 0 LSB up to about 1600 LSB for the AVATAR-I data set in about 7 km altitude. The mean signal levels range from about 100 LSB to 500 LSB for both data sets. Furthermore, it is interesting to realize that no





Mie-cloudy winds are available for the AVATAR-T data set between $1\,\mathrm{km}$ and $4.5\,\mathrm{km}$ altitude which represents the dust-laden

SAL. Thus, Mie-cloudy winds are indeed just retrieved from cloud returns and not from aerosol-rich regions. Outliers appear in all altitudes, but mainly for low signal levels.

The altitude-dependent EE is depicted in Fig.10 (b). In general, as for Rayleigh-clear winds, the EE is smaller in regions of larger signal levels. However, as the Mie-cloudy EE does not directly depend on the signal level but on the accuracy of the actual fit-routine, this behavior is less pronounced. The mean EE ranges from about $1\,\mathrm{ms}^{-1}$ to $3\,\mathrm{ms}^{-1}$ for both data sets.

The altitude dependent Aeolus error wrt to the 2-$\mu$m DWL observations is shown in Fig. 10 (c). From the AVATAR-I data it can be seen that the mean error is obviously negative ($\approx -2.5\,\mathrm{ms}^{-1}$) for altitudes between $4.5\,\mathrm{km}$ and ground where the random error is relatively small. This is also the region with larger scattering ratios as they are shown in Fig. 10 (d) and thus confirms that the accuracy and precison of Mie-cloudy winds depends on the scattering ratio (see also Fig. 9 (d)). The AVATAR-T data set provides not enough data points to draw the same conclusion. The scattering ratio in general varies from

close to 1 up to 100, whereas the mean scattering ratio varies from close to 1 up to 30 for both data sets. For the AVATAR-I data set, larger scattering ratio values up to 30 are prominent in altitudes from ground up to $4.5\,\mathrm{km}$ which is also the region that shows an enhanced negative systematic error.

The altitude dependency of the measured 2-$\mu$m DWL winds is shown in Fig. 10 (d). The wind speed range is much smaller than for Rayleigh-clear winds and varies from about $-30\,\mathrm{ms}^{-1}$ to $50\,\mathrm{ms}^{-1}$ for AVATAR-I and from about $-20\,\mathrm{ms}^{-1}$ to

$15\,\mathrm{ms}^{-1}$ for AVATAR-T. As for Rayleigh-clear winds, the highest wind speeds are found in the vicinity of the jet stream in the AVATAR-I data set.

## 7  Summary

In the past three years, DLR strongly contributed to Aeolus CalVal activities by means of airborne wind lidar measurements. In this study, the data quality of Aeolus L2B wind products in two regions of particular interest to NWP, namely the North

Atlantic jet stream region and the region of tropical winds affected by dust transport from the Sahara, are analyzed. This analysis is based on airborne wind lidar data acquired from DLR's Falcon aircraft during two airborne campaigns performed in Iceland (AVATAR-I) and Cape Verde (AVATAR-T). During the AVATAR-I campaign, conducted from Keflavik, Iceland in September 2019, 10 satellite underflights on ascending and for the first time descending orbits were performed. During the AVATAR-T campaign, conducted from Sal, Cape Verde in September 2021, 11 satellite underflights on ascending and

descending orbits were executed. In total, these underflights lead to about $19000\,\mathrm{km}$ along the Aeolus measurement track that are used for comparison.

Based on a statistical analysis, the systematic and random errors of Aeolus HLOS wind observations are determined by comparing to 2-$\mu$m DWL observations. The 2-$\mu$m DWL is suitable as a reference instrument due to the low systematic and random errors that come along with the heterodyne detection measurement principle of the system. This way, reliable values

for the systematic and random errors for the AVATAR-I data set are determined to be $(-0.8\pm0.2)\,\mathrm{ms}^{-1}$ and $5.5\,\mathrm{ms}^{-1}$ for Rayleigh-clear winds, and $(-0.9\pm0.1)\,\mathrm{ms}^{-1}$ and $2.7\,\mathrm{ms}^{-1}$ for Mie-cloudy winds, respectively. For the AVATAR-T data set,



the systematic and random errors are $(-0.1 \pm 0.3)\,\mathrm{m\,s^{-1}}$ and $7.1\,\mathrm{m\,s^{-1}}$ for Rayleigh-clear winds, and $(-0.7 \pm 0.2)\,\mathrm{m\,s^{-1}}$ and $2.9\,\mathrm{m\,s^{-1}}$ for Mie-cloudy winds, respectively. Thus, within the given uncertainty, the systematic error fulfills the requirement of being below $0.7\,\mathrm{m\,s^{-1}}$ for both wind products and both campaign data sets. This confirms the sucessful correction schemes for

systematic errors that have been identified in the early phase of the mission. Hot pixels and the thermal variations on the Aeolus telescope mirror are treated in the refined Aeolus data processor after the release of the processor baselines 10 and following. The random error of Rayleigh-clear winds is significantly larger than specified ($2.5\,\mathrm{m\,s^{-1}}$), which is due to the overall lower signal levels most likely caused by a combination of instumental misalignment, the wavefront error of the $1.5\,\mathrm{m}$ telescope and laser induced contamination. The random error of Mie-cloudy winds is close to the specifications.

The results are confirmed by comparison against in-situ data from the Falcon nose-boom which yield a systematic and random error for Rayleigh-clear winds of $(-0.3 \pm 0.5)\,\mathrm{m\,s^{-1}}$ and $5.0\,\mathrm{m\,s^{-1}}$ for the AVATAR-I data set and $(-0.2 \pm 0.4)\,\mathrm{m\,s^{-1}}$ and $4.6\,\mathrm{m\,s^{-1}}$. The lower random error compared to the $2$-$\mu$m DWL analysis determined for the AVATAR-T data is shown to be due to the altitude-dependency of the random error which is caused by the height-dependency of the signal levels, especially in aerosol-laden regions.

A detailed analysis of the Rayleigh-clear wind errors reveals that they are neither depending on the actual wind speed, nor the geolocation (latitude), the time difference between $2$-$\mu$m DWL observation and satellite overflight and also not on the scattering ratio, which further confirms a proper calibration scheme of the Aeolus instrument. Moreover, based on an altitude-dependent analysis of the Aeolus wind speed error, it is shown that the random error mainly depends on the signal levels and that it is well represented by the estimated error, assumed that a proper quality control of Aeolus data by means of appropriate

EE thresholds together with an additional Z-score-based outlier removal is performed in advance.

The detailed analysis of Mie-cloudy wind errors demonstrated that they are also not depending on the actual wind speed, the geolocation (latitude) and the time difference between $2$-$\mu$m DWL observation and satellite overflight, but show a dependence on the scattering ratio. In particular, the systematic error drifts to more nagative values and the random error reduces for larger scattering ratios. This is also confirmed by the altitude-dependent analysis that shows larger negative systematic errors

in altitudes where the scattering ratio is enhanced. Furthermore, it is revelaed that Mie winds are indeed only available from cloudy returns. Aerosol-laden regions as the SAL only provide Rayleigh-clear and no Mie winds.

This analysis is an important contribution to evaluate the quality of Aolus winds and the fullfilment of mission requirements defined in advance. It shows, that Aeolus the Aeolus (alomst) fullfills the required values that were originally specified. The larger random errors are due to the lower signal levels that are caused by a combination of initial misalignment and laser

induced contamination.

*Author contributions.* Benjamin Witschas prepared the main part of the paper manuscript and performed the corresponding analyses. Stephan Rahm performed the $2$-$\mu$m DWL data analysis. Christian Lemmerz, Oliver Lux and Uwe Marksteiner provided A2D data for further comparison and helped with the preparation of the manuscript. Alexander Geiß and Andreas Schäfler performed the weather forecast during the campaigns, provided meteorological data for the data analyses and helped with the preparation of the paper manuscript. Fabian Weiler



provided the original and reprocessed Aeolus data that is used within this study. Oliver Reitebuch helped with the data analyses and the preparation of the paper manuscript.

*Competing interests.* The authors declare that they have no conflict of interest.

*Acknowledgements.* The presented work includes preliminary data (not fully calibrated/validated and not yet publicly released) of the Aeolus mission that is part of the European Space Agency (ESA) Earth Explorer Programme. The processor development, improvement and product
reprocessing preparation are performed by the Aeolus DISC (Data, Innovation and Science Cluster), which involves DLR, DoRIT, ECMWF, KNMI, CNRS, S&T, ABB and Serco, in close cooperation with the Aeolus PDGS (Payload Data Ground Segment). The analysis has been performed in the frame of the Aeolus Data Innovation and Science Cluster (Aeolus DISC). We are grateful to our ESA colleagues Thorsten Fehr (Aeolus scientific campaign coordinator) and Jonas von Bismarck (Aeolus data quality manager), for their support of the study. Moreover, we would like to thank the DLR flight experiments department for the realization of the AVATAR-T airborne campaign
despite the obstacles posed by the COVID-19 pandemic.



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
