# Peer review of "Validation of the Aeolus L2B wind product with airborne wind lidar measurements in the polar North Atlantic region and in the tropics"

_Atmospheric Measurement Techniques, 2022_

## Author Comment (AC1)

**(Author response)**

**Overview:**
This paper presents results from two DLR flight campaigns in the frame of the Aeolus Calibration and Validation activities. Beyond the presentation of the data, the analysis focuses on the validation of Aeolus observations. The method is clearly explained and limitations of the analysis are properly acknowledged. I only have minor comments on this paper.

**Specific comments:**
- line 351: Could you think of a reason why the later dataset (AVATAR-T) has a lower EE threshold than the earlier one, when the signal quality was higher?
  For the Mie-cloudy winds, the EE threshold was determined to be 5.5 m/s for the AVATAR-I campaign and 5.0 m/s for the AVATAR-T campaign, hence indeed lower for AVATAR-T. The EE of Mie-cloudy winds depends on the signal level or rather the quality of the fit that is used in the processor which is not necessarily proportional to the laser pulse energy but to the backscattering cross-section of the aerosols and clouds. As can be seen from Fig. 10, a, the useful signal for Mie-cloudy winds was actually larger for the AVATAR-T campaign. Thus, a slightly smaller EE threshold is reasonable. Furthermore, as can be seen from Fig. 4, c, neither the number of available data points nor the statistical parameters (mean, STD, scaled MAD) change significantly between an EE threshold of 5.0 m/s and 7.0 m/s for this particular data set. For further clarification, we added the following sentence to the manuscript:
  - Line 352: It is worth mentioning that the statistical comparison of AVATAR-T Mie-cloudy winds is not very sensitive to the actual EE threshold. For instance, an EE threshold of up to 7.0 m/s would yield similar results.
- line 362: Is it visible somewhere that the number of colocations is small for the last 6 flights? You could point to where that can be seen, or mention that it is "not shown".
  Thanks a lot for this hint. The information on available Rayleigh-clear and Mie-cloudy data points for each research flight that could be used for comparison is given in Table 2 (last two columns). Thus, we added this information to the manuscript and additionally point to the degrading performance of the 2-μm DWL during the last 5 research flights of the AVATAR-T campaign:
  - Line 148: Additionally, the number of Aeolus observations that could be validated by the 2-μm DWL is given for both Rayleigh-clear and Mie cloudy winds. It can be seen that the 2-μm DWL was degrading during the AVATAR-T campaign, leading to very low data coverage for the last six research flights (see also Sect. 5). The uncertainties of the 2-μm DWL observations are however not affected thanks to the heterodyne-detection measurement principle.
  - Line 362: The decreased number of data points observed during the last six AVATAR-T underflights (for detailed itemization see Table 2) was due to the fact that the 2-μm DWL was degrading during the campaign period.

- line 387: random error of the 2micron lidar: You give only one value and do not discuss it much. Is the random error stable in time? You mentioned some loss of signal during the last flights, could this affect the random error for instance?

  In Witschas et al., 2020 the systematic and random error of 2-µm DWL observations is characterized by comparisons to dropsonde measurements that have been performed within the last two decades. These comparisons demonstrate that the 2-µm errors are rather stable in time and that the random error is always of the order of 1 m/s. As the 2-µm DWL is based on a heterodyne-detection principle, the random error is not proportional to the signal level as it is true for direct-detection wind lidar systems. Whenever the wind can be retrieved from the power spectrum, it has a similar quality. Otherwise, no wind can be determined. Thus, a worse 2-µm DWL performance leads to significantly lower data coverage, but not to an increase in the systematic or random error. This is also the reason why almost no wind observations are available from the last research flights of the AVATAR-T campaign. Regarding this, we added the following sentences to the manuscript for further clarification:
    - Line 226: The systematic error of horizontal wind speed is determined to be below 0.1 m/s and the corresponding random error varies between 0.9 m/s and 1.5 m/s, whereas these errors are composed of the contribution of the 2-µm DWL and the dropsonds as well as the corresponding representativeness errors. Thus, the random error of the 2-µm DWL can be considered to be of the order of 1 m/s.
    - Line 372: It is worth mentioning here that due to the heterodyne-detection measurement principle of the 2-µm DWL, the progressive misalignment only led to a reduction in the data coverage but not to an increase of the systematic or random error of the wind observations over the course of the campaign.
    - Line 387: The impact of the 2-µm DWL uncertainty of about 1 m/s (see also Witschas et al. (2020)) on the determined random error is only marginal.
- lines 406-407: Rayleigh-clear winds in aerosol-loaded parts of the sky: can you provide a quick explanation of why there are valid Rayleigh winds with high scattering ratio (SR)? Is it possible to estimate how much these winds are affected by the aerosols? (e.g. Can the additional error due to the high scattering ratio be quantified?) Or could you maybe give a reference that further discusses this point? I realize that some Rayleigh winds with high SR are shown in fig. 7, but I am not really convinced that the errors are negligible because of the small number of points above SR=4.
- This is a very good point. Despite the long-term effort of the Aeolus DISC team to improve the determination of the scattering ratio for the purpose of an appropriate clear/cloudy classification, the scattering ratio data still has to be considered preliminary, especially for Rayleigh-clear winds. Currently, and for the processor versions used in this study, the scattering ratio is determined from Mie-channel data on measurement level in the L1B processor. This information is afterwards used to classify the winds as "cloudy" and "clear". In the next step, the L1B scattering ratio data is copied to the L2B, where the Rayleigh data is given on observational level, i.e. 87 km horizontal resolution. In broken cloud conditions, it may happen that valid Rayleigh-clear winds are retrieved based on a sufficient number of measurements during one observation. However, the scattering ratio, which is calculated from the Mie-channel can still be affected by e.g. a strong cloud return which leads to a mean scattering ratio significantly larger than 1. In the L2B processor itself, no scattering ratio threshold is used. Thus, all Rayleigh-clear observations that are principally based on Rayleigh-clear measurements lead to valid L2B data. Hence, in the current state, the retrieved scattering ratio for Rayleigh-clear winds is not necessarily reliable. Still, we think it is useful to show the data, as it reflects the current status of the processor and that there is no significant systematic or random error dependency on the scattering ratio, although it is agreed that the number of

available data points is sparse. To further clarify this situation, we added the following paragraph at the beginning of section 6 (Aeolus error dependency):

- o Further, it is verified whether the error has any dependency on the scattering ratio, which might be induced by a cross-talk between the signals from the Rayleigh and the Mie channel, respectively. It has to be mentioned that the Aeolus scattering ratio data is still preliminary and subject to processor improvements. For instance, for the AVATAR-I data set, the scattering ratio values were taken from the L2B data. For the AVATAR-T data set, however, scattering ratio values were partly set to 1 to avoid problems with the assimilation of Aeolus data in the ECMWF model. Hence, the scattering ratio values were calculated from L1B after adaptation to the L2B grid and averaging as it was done for the wind processing. Furthermore, it has to be pointed out that the scattering ratio values retrieved for Rayleigh-clear winds in broken-cloud conditions could be faulty due to grouping issues that appear when going from L1B to L2B data. Thus, especially for Rayleigh-clear winds, no solid conclusions can be drawn from scattering ratio data. Nevertheless, the data is shown to represent the current status of the Aeolus data processor.

- line 464: "confirming that Aeolus calibration is working properly": the airborne lidar dataset provides a validation of Aeolus, and it's good news that they agree! But we also know some limitations of the calibration do appear in other analyses. Can you discuss the precision of the current analysis? The sensitivity is probably limited by the relatively small number of points, and because of that, smaller calibration imprecisions cannot be detected. I am thinking about the bias depending on atmospheric temperature, for instance. If you could give an estimation of the possible range of undetected calibration errors, it would be even better!

- This is a good point and addresses the limitations of each reference data set. Thus, it is even more important that different data sets in different parts of the world are available and used for CalVal activities. The mentioned height- or rather temperature dependency that is seen by means of ECMWF comparisons can indeed not be reproduced by our airborne measurements. This is due to the rather low number of data points, but also the limited geographical extent of the performed measurements. In order to mention these limitations, we added the following sentence after line 465:

- o It has to be mentioned that these results are restricted to certain geographical regions, certain time periods, and a limited number of data points. Thus, probably not all error contributions can be detected in this analysis, especially if those are related to strong non-periodic sources like strong deviations from the atmospheric temperature profile or orbital variations. For an even more conclusive error characterization, data from other CalVal teams as well as model data is needed.

- lines 538-540: "No Mie winds are available in the SAL": Do you have a hypothesis to explain why there are no Mie winds from the SAL layer? (e.g. depolarization is too strong, accumulation length in the L2B processor is too short?)

- It is known since the start of the mission that most of the winds retrieved from the Mie channel are originating from cloud returns and not from aerosol returns. This is explained by the scattering ratio thresholds and the grouping schemes that are used in the Aeolus processor. Thus, for the shown data set, the backscatter from the SAL region has either too-low signal levels, and with that, a too-low SNR, or the signal for the SAL region is even blocked by clouds that can originate on the top of the SAL layer. The strong signal attenuation within the SAL layer during the AVATAR-T campaign can, for instance, be seen from the Rayleigh-clear mean useful signal, as shown in Fig. 8. On the contrary, from the measurements performed by the ALADIN airborne demonstrator (A2D) during the AVATAR-T campaign, a very large number of valid Mie-cloudy winds is retrieved in the SAL region. Based on these data, investigations are ongoing to adapt the scattering ratio

thresholds of Mie-winds as well as the QC schemes for filtering gross errors whilst retaining good-quality winds from the aerosol layer. Further improvement of the Mie data coverage may be additionally achieved by refining the fit routines and background corrections in the Mie wind retrieval. For further clarification, the following sentence is added to the manuscript:

    o   Line 540: This is a known issue and is related to the scattering ratio thresholds and grouping schemes that are currently used in the Aeolus processor.

**Grammar, phrasing, typos:**

- line 89-90: "It can be seen that most of the CalVal activities using observations as a reference to determine the systematic and random errors in specific geographical regions for random wind situations above the measurement sites." I didn't understand this sentence.
  The sentence was deleted and the subsequent sentence was adapted according to:" In this paper, the aforementioned work of other CalVal teams is extended by the analysis of the L2B wind quality in two dedicated regions over the North Atlantic:…".
- line 92 "whereas"?
  Changed to "where"
- line 161 replace "time-resolved" by "In a time-resolved manner"?
  Adopted
- line 176 precise "any differences in atmospheric conditions between instrument response calibration and wind observation"?
  For further clarification, the sentence was extended according to:" any differences of both quantities between the geographical location of the instrument response calibration and the one of the actual wind observation have to be taken into account.".
- line 297 "the the"
  Adopted
- lines 343-344 Do you mean that the difference between both quantities start to increase?
  Yes, indeed. The sentence was adapted to:" For even larger EE thresholds, the difference of the scaled MAD calculated without (red) and with (magenta) outliers starts to increase, indicating that the outliers start to have an impact on the calculated statistical parameters.".
- line 423: "altituds"
  Changed to altitudes.
- lines 443-445: Wouldn't it be more readable if the numbers were presented in a table instead?#
  As the averaging intervals are not particularly interesting and rather "artificially" chosen, it is preferred to just mention them in the text and not produce an extra table.
- line 490: "where is goes" -> "where it goes"
  Adapted
- line 500: "up 10 and higher" -> "up to 10 and higher"? legend figure 10: "AVATAR-E" -> "AVATAR-T"
  Both adapted
- line 549: "provides not enough" should be rephrased into "does not provide enough"
  Done
- line 598: "that Aeolus the Aeolus (alomst)"
  The sentence was adapted to:" It shows, that Aeolus (almost) fulfills the required values that were originally specified.".

---

## Author Comment (AC2)

**(Author response)**

**Referee #2**: The results of two airborne campaigns (AVATAR-I and AVATAR-T) for the Aeolus validation are analyzed in detail in this article. In the validation process, quality control of Aeolus data is critical. In this paper, the authors improve the Z-score method using median and scaled MAD to avoid the impact of a single outlier. Based on a statistical analysis, the systematic and random errors of Aeolus HLOS wind observations are determined by comparing to 2-μm DWL observations. A detailed analysis of the Rayleighclear Mie-cloudy wind errors is carried out. This analysis is an important contribution to evaluate the quality of Aolus winds and the fulfillment of mission requirements defined in advance. This paper has feasible data, proper method and detailed data analysis. Thus, this paper is recommended to be published as it is.